# CODA: Temporal Domain Generalization via Concept Drift Simulator

## Abstract

In real-world applications, machine learning models are often notoriously blamed for performance degradation due to data distribution shifts. Temporal domain generalization aims to learn models that can adapt to "concept drift" over time and perform well in the near future. To the best of our knowledge, existing works rely on model extrapolation enhancement or models with dynamic parameters to achieve temporal generalization. However, these model-centric training strategies may involve the *unnecessarily comprehensive* interaction between data and model to train the model for distribution shift, accordingly.[1] To this end, we aim to tackle the concept drift problem from a data-centric perspective and naturally bypass the cumbersome interaction between data and model. Developing the data-centric framework involves two challenges: (i) existing generative models struggle to generate future data with natural evolution, and (ii) directly capturing the temporal trends of data with high precision is daunting [2]. To tackle these challenges, we propose the COncept Drift simulAtor (CODA) framework incorporating a predicted feature correlation matrix to simulate future data for model training. Specifically, the feature correlations matrix serves as a delegation to represent data characteristics at each time point and the trigger for future data generation. Experimental results demonstrate that using CODA-generated data as training input effectively achieves temporal domain generalization across different model architectures with great transferability. Our source code is available at: `https://anonymous.4open.science/r/coda-D648`

## 1 Introduction

The remarkable progress in machine learning has spurred many applications in real-world scenarios (Wei et al., 2019; Tan et al., 2020; Strudel et al., 2021; Cheng et al., 2021; Chang et al., 2021; Dai et al., 2021; Joseph et al., 2021; Jiang et al., 2021). Typically, the historical training data is assumed with the same distribution as future data. However, this assumption is not usually satisfied in real-world settings due to practical data evolving. In other words, the trained model may show performance degradation due to the temporal distribution drift between the training and near-future testing data. Distribution drift hereby poses substantial challenges for researchers to perfectly exploit data on model training. In this manner, Domain Adaptation (DA) (Lu et al., 2020; Liu et al., 2020; Balaji et al., 2019; Kang et al., 2019; Liu et al., 2021), Domain Generalization (DG) (Xu et al., 2020; Yan et al., 2020; Qiao et al., 2020; Chattopadhyay et al., 2020; Piratla et al., 2020; Cha et al., 2022), and Temporal Domain Generalization (TDG) (Bai et al., 2022; Nasery et al., 2021) are proposed to address the distribution shift with different settings.

In contrast to DA, which requires access to unlabeled data in the target domain, DG and TDG are more practical in real-world scenarios where feature information about the target domain is typically unavailable. Existing DG methods aim to improve generalization ability across "distinct" categorical domains, such as different datasets and sources. However, in real-world applications, data may naturally evolve over time, leading to the emergence of temporal domains. For instance, in predicting seasonal flu trends using Twitter data (Bai et al., 2022), as the platform gains more

---

[1] The fundamental cause in concept drift arises *only* from data perspective.

[2] Directly predicting a whole dataset requires data distribution estimation, leading to prohibitive computational costs. (Please see Section 3.1 for more details.)

users, forms new connections, and experiences demographic shifts, the correlation between user profiles and flu predictions changes, leading to outdated models. Therefore, temporal distribution shift, a commonly observed phenomenon in the real world, brings significant changes in the joint distribution of features and labels with smooth data shift over time, known as "concept drift". While DA and DG methods are designed to bridge the gap between different data distributions, they do not consider the *chronological* domain indices and smooth drift with the underlying pattern. Such a unique challenge of temporal distribution shift requires further research towards TDG.

To alleviate concept drift for achieving TDG, Gradient Interpolation (GI) (Nasery et al., 2021) and DRAIN (Bai et al., 2022) introduced model-centric strategies, either incorporating specialized regularization or parameter forecasting techniques, and may involve unnecessary interaction between data and model. Noticed that the temporal distribution shift issue is purely from a data perspective, a natural question is raised: ***Can we achieve TDG via a data-centric approach without involving data-model interaction?*** Our pioneer research aims to address concept drift from a novel data-centric angle rather than against the parallel model-centric strategy.

We provide a positive answer to this question and propose a data-centric approach to simulate future data via out-of-distribution (OOD) data generation. However, it is non-trivial to generate OOD data reflecting the concept drifts from historical data. A naive approach is to directly capture the temporal trends to predict future data using the sequential model. However, this approach is demonstrated to be problematic in our preliminary analysis (see Section 3.1). Specifically, we leverage an LSTM unit for simulating future samples, where the model is trained with dataset-level Kullback-Leibler (KL) divergence. The results demonstrate poor performance in the TDG setting due to the limited model prediction capacity. To further develop an effective data-centric framework for future data generation, there are two main challenges: (i) Generating out-of-distribution future data is beyond the capacity of many existing generative models. Most generative models aim to generate the data following the same distribution of observed training data and therefore are not able to extrapolate the future data distribution. (ii) Directly capturing the temporal trends along chronological source domains with high precision is daunting (the arduousness is demonstrated in Section 3.1).

To tackle the aforementioned challenges, we propose COncept Drift simulAtor (CODA), a data simulation framework incorporating feature correlation matrices to capture temporal trends within the historical source domains for generating future data. Specifically, CODA consists of two steps: (i) Extracting a dynamic feature correlation matrix from source domains, which is utilized to learn an informative one for the upcoming domain, and (ii) Simulating future training data based on the predicted future feature correlation. Subsequently, the generated future data can be utilized as the training data for prediction models, such as MLP-based or tree-based classifiers, enhancing the performance on the unseen future domain. Theoretical analysis ensures that under practical assumptions, the predicted correlation is dependable. Experimental results demonstrate the effectiveness of CODA in TDG by providing high-quality simulated data. Our contributions are as follows:

- CODA designs a novel model-agnostic approach to tackle the issue of concept drift. We design the feature correlation matrix as a data delegation and future data generation trigger to enhance OOD generation quality.

- Theoretical and experimental analysis indicates that CODA can be facilitated for future data generation by considering the temporal trends of feature correlations.

- Experimental results demonstrate that the simulated future dataset can be leveraged as training data with transferability for various model architectures.

## 2 PRELIMINARY

### 2.1 TEMPORAL DOMAIN GENERALIZATION PROBLEM FORMULATION

We consider temporal prediction tasks with evolved data distribution. In the training stage, given $T$ observed continuous source domains $\mathcal{D}_1, \mathcal{D}_2, \ldots, \mathcal{D}_T$, the data instance within domain $\mathcal{D}_t$, sampling from distributions at distinct time points $t$, is defined as $\{(x_i^t, y_i^t) \in \mathcal{X}_t \times \mathcal{Y}_t\}_{i=1}^{N_t}$, where domain index $t = 1, 2, \ldots, T$. Here, $x_i^t$ and $y_i^t$ represent the input features and labels, respectively, $N_t$ denotes the sample size, and $\mathcal{X}_t \times \mathcal{Y}_t$ signifies the feature and label spaces at time $t$. The ultimate goal is to train a model generalized well on some target (test) domain *in the future*, i.e., $\mathcal{D}_{T+1}$. In the temporal

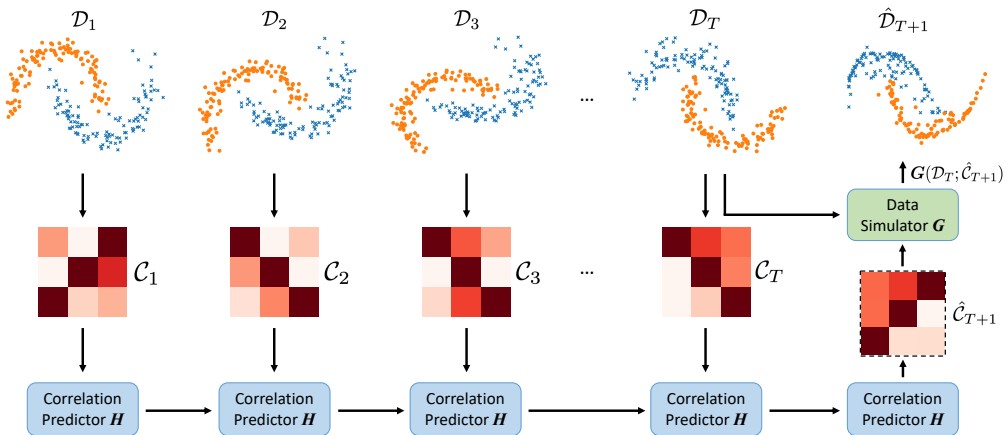

Figure 1: An overview of the proposed framework CODA, consisting of a Correlation Predictor (see Section 3.2) and a Data Simulator (see Section 3.3). In the first stage, the Correlation Predictor is trained on feature correlation matrices at each time point $\mathcal{C}_i$. In the second stage, the predicted correlation matrix $\hat{\mathcal{C}}_{T+1}$ served as prior knowledge for Data Simulator $G$ to be updated by Equation 5.

domain generalization problem, the concept drift across different domains is commonly assumed, i.e., the data distribution smoothly evolved across time (domain index) following underlying but unknown patterns. The notations are summarized in Table 4 in the Appendix.

## 2.2 RELATED WORKS

**Domain Generalization (DG).** The goal of DG is to enhance model generalization ability to the open *unordered* target domains given multiple domains data without accessing target domains. Several methods have been proposed for this task in recent years (Muandet et al., 2013; Cha et al., 2022; Chattopadhyay et al., 2020; Qiao et al., 2020; Motiian et al., 2017; Nasery et al., 2021; Dou et al., 2019). Existing DG methods can be categorized into three branches (Wang et al., 2022): (1) Data manipulation; The input data can be manipulated to learn a general representation via generating diverse data, including data augmentation (e.g., randomization, and transformation) (Tobin et al., 2017; Tremblay et al., 2018) and diverse data generation (Liu et al., 2018; Qiao et al., 2020). (2) Representation learning. The generalized representation learning can be pursued by domain-invariant property via adversarial training, invariant risk minimization, or feature alignments across domains (Ganin et al., 2016; Gong et al., 2019). Another way is feature disentanglement with domain-shared and domain-specific features for better generalization (Li et al., 2017). (3) General learning strategy. The learning strategy for generalization consists of ensemble learning (Mancini et al., 2018), meta-learning (Dou et al., 2019), and gradient operation (Huang et al., 2020), et al.

**Temporal Domain Generalization (TDG).** TDG is a promising yet challenging problem. The difference with DG (categorical domain index) falls in the *ordered* (continuous) time domain index and thus meticulously requires distribution evolving pattern capture. Unfortunately, there are only a few existing model-centric works on TDG. E.g., Gradient Interpolation (GI) (Nasery et al., 2021) explicitly chases model extrapolation to the near future via the first-order Taylor expansion in the designed loss function; Drift-Aware Dynamic Neural Network (DRAIN) (Bai et al., 2022) formulates a Bayesian probability framework to represent the concept drift over domains and build a recurrent graph to capture dynamic model parameters for the temporal shift. These methods are model-centric and counter the temporal data shift via enhancing model extrapolation or generalization ability or adjusting model parameters accordingly. However, the model may be infeasible to counter all possible future temporal drifts due to limited model capacity. To this end, we proposed a brand-new model-agnostic approach, named COncept Drift simulAtor (CODA), for achieving TDG via out-of-distribution (OOD) generation. In this way, the future data can be simulated and then leveraged into model training to enhance generalization. The data-centric approach CODA is model-agnostic with transferability, i.e., you only simulate once for any possible model architectures.

## 3 CONCEPT DRIFT SIMULATOR (CODA)

This section presents a comprehensive overview of our proposed CODA framework, as depicted in Figure 1. Specifically, starting with the feature correlation matrices at various time points, CODA identifies evolving trends to extrapolate the feature correlation matrix for future domains. Leveraging the predicted correlation matrix, CODA subsequently synthesizes future data for model training. The details and pseudo-code of the CODA workflow can be found in Appendix B.

### 3.1 LIMITATION OF DYNAMICALLY GENERATING FUTURE DATA

To tackle the challenge of the concept drifts through a data-centric lens, our goal is to model the underlying distribution shift by capturing temporal trends within historical datasets. A straightforward approach is to employ a naive RNN-based architecture for learning temporal patterns from chronological datasets, named Prelim-LSTM. Given $T$ source domains from the Elec2 dataset, $\mathcal{D}_1, \mathcal{D}_2, \ldots, \mathcal{D}_T$, we employ an LSTM unit to generate the synthetic future domain $\hat{\mathcal{D}}_{T+1}$. To capture the temporal trend of the probability distribution with limited samples at each time point, we employ Kullback-Leibler (KL) divergence as the objective loss. Specifically, we first approximate the conditional distributions $\mathcal{P}(\mathcal{X}_t|\mathcal{Y}_t)$ for both the predicted and ground truth features via Kernel Density Estimation (KDE). Utilizing the prior probability of labels $\mathcal{P}(\mathcal{Y})$, we proceed to compute the joint distribution probability $\mathcal{P}(\mathcal{X}_t, \mathcal{Y}_t)$ between all features and labels. In each training time point, the KL-divergence loss is calculated between the approximated joint distributions of the predicted and ground truth datasets across all features. The objective function is formulated as:

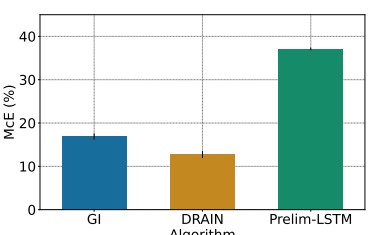

Figure 2: Misclassification Error (McE) rates comparison of the preliminary experiment, where we compare the preliminary data-centric setting, Prelim-LSTM, to the two model-specific SOTAs.

$$\mathcal{L}_{KL} = \sum_{i=1}^{\mathcal{I}} L_{KL}^i[\mathcal{P}(\mathcal{X}_t^i, \mathcal{Y}_t)_{\text{pred}} \| \mathcal{P}(\mathcal{X}_t^i, \mathcal{Y}_t)_{\text{gt}}], \tag{1}$$

where $L_{KL}^i$ denotes the KL-divergence of the $i^{th}$ feature, $\mathcal{P}(\mathcal{X}_t^i, \mathcal{Y}_t)_{\text{pred}}$ represents the predicted joint distribution between $i^{th}$ feature and the labels, and $\mathcal{P}(\mathcal{X}_t^i, \mathcal{Y}_t)_{\text{gt}}$ is the joint distribution of the ground truth dataset for the same feature. The objective is to minimize $\mathcal{L}_{KL}$ to align the predicted joint distribution of each feature as closely as possible with the ground truth. Ideally, given the distribution at the latest observed time point $\mathcal{D}_T$, the trained LSTM model tends to predict the pseudo future data distribution $\hat{\mathcal{D}}_{T+1}$, leveraging the temporal relationships among $\mathcal{D}_1$ to $\mathcal{D}_T$.

However, our preliminary experiments demonstrate that it is difficult to directly capture the temporal trends of underlying joint distribution between $\mathcal{X}$ and $\mathcal{Y}$ through chronological data. As shown in Figure 2, despite independently approximating the KDE of each feature to the ground truth distribution, the MLP trained on the data predicted by the Prelim-LSTM falls to achieve comparable performance to the two model-specific SOTAs, GI (Nasery et al., 2021) and DRAIN (Bai et al., 2022). It also indicates the necessity of considering feature correlations for capturing concept drift data distribution. Ideally, one would consider all features $\mathcal{X}$ concurrently to achieve a more accurate approximation of the data distribution at each temporal instance. However, this becomes computationally infeasible in high-dimensional spaces. To be more specific, grid sampling in an $n$-dimensional space incurs a time complexity of $O(k^n)$, where $k$ represents the number of grid points along each dimension. As $n$ increases, the computational cost becomes prohibitive. Therefore, we propose CODA to tackle the aforementioned challenges by breaking down the future distribution generation process into two manageable components and addressing each in a sequential manner.

### 3.2 CORRELATION PREDICTOR UNDER TEMPORAL TRENDS

Instead of directly predicting the data distribution for future domains, CODA begins with the Correlation Predictor component to capture the temporal trends of concept drift, as shown in Figure 1. In the first stage of CODA, the Correlation Predictor is specifically designed to discern evolving

trends in feature correlations across chronological time domains. Given the observed source domains $\mathcal{D}_1, \mathcal{D}_2, \ldots, \mathcal{D}_T$, we extract the feature correlation matrices from each domain to serve as meta-data. These matrices can be computed using either traditional statistical approaches, such as Pearson Correlation (Pearson, 1895), or learning-based methods like Self-attention mechanisms (Song et al., 2019) and Graph-based Adjacency Matrix learning (Liu et al., 2022). Armed with these chronological feature correlation matrices, the Correlation Predictor $\boldsymbol{H}(\cdot)$ is trained to capture the evolving trends in correlation matrices under the influence of concept drift:

$$\hat{\mathcal{C}}_{T+1} = \boldsymbol{H}(\mathcal{C}_1 \ldots \mathcal{C}_T). \tag{2}$$

The objective loss of the Correlation Predictor $\mathcal{L}_{CP}$ is formulated as follows:

$$\mathcal{L}_{CP} = \sum_{t=1}^{T} \left( \|\hat{\mathcal{C}}_t, \mathcal{C}_t\|_2 + \|\hat{\mathcal{C}}_t, \mathcal{C}_t\|_1 + \lambda_{CE} \mathcal{L}_{CE}(\hat{\mathcal{C}}_t, \mathcal{C}_t) \right), \tag{3}$$

where $\| \cdot \|_2$ and $\| \cdot \|_1$ are the $l_2$-norm and $l_1$-norm regularization, $\lambda_{CE}$ denotes the weight of $\mathcal{L}_{CE}$, and $\mathcal{L}_{CE}$ is the Cross Entropy loss between the predicted correlation matrix $\hat{\mathcal{C}}_t$ and the ground truth feature correlation matrix $\mathcal{C}_t$. It's worth noting that the Correlation Predictor component is modular and can be implemented or substituted with any RNN-based neural network architecture.

## 3.3 DATA SIMULATOR FOR CONCEPT DRIFT DATA GENERATION

Upon completion of training the Correlation Predictor $\boldsymbol{H}(\cdot)$, the second stage of CODA utilizes the feature correlation matrix $\hat{\mathcal{C}}_{T+1}$ estimated by $\boldsymbol{H}(\cdot)$ for the unseen future domain simulation. This estimated correlation matrix $\hat{\mathcal{C}}_{T+1}$ serves as a prior knowledge in the Data Simulator $\boldsymbol{G}(\cdot)$ to predict the future domain dataset $\hat{\mathcal{D}}_{T+1}$:

$$\hat{\mathcal{D}}_{T+1} = \boldsymbol{G}(\mathcal{D}_T; \hat{\mathcal{C}}_{T+1} | \theta_G). \tag{4}$$

By incorporating $\hat{\mathcal{C}}_{T+1}$ and aligning with the data distribution of the current domain $\mathcal{D}_T$, $\boldsymbol{G}(\cdot)$ aims to approximate both the observed data distribution and the underlying feature correlations. The objective function of the Data Simulator $\boldsymbol{G}(\cdot)$ is updated by minimizing the following loss:

$$\begin{aligned} \mathcal{L}_G &= ELBO(\mathcal{D}_T, \hat{\mathcal{D}}_{T+1}; \theta_G) + \lambda_C \mathcal{R}_C(\hat{\mathcal{C}}_{T+1}) \\ &= \mathbb{E}_{q(z|(x,y))}[\ln p_\theta((x,y)|z; \theta_G)] - \mathcal{L}_{KL}[q(z|(x,y))||p(z)] + \lambda_C \|(\hat{\mathcal{C}}_G - \hat{\mathcal{C}}_{T+1})\|_1, \end{aligned} \tag{5}$$

where $ELBO$ represents the classic reconstruction loss, $\hat{\mathcal{C}}_G$ denotes the correlation matrix of the CODA-generated dataset, $\mathcal{R}_C$ serves as a regularization term that ensures $\boldsymbol{G}(\cdot)$ following the prior predicted feature correlation matrix $\hat{\mathcal{C}}_{T+1}$, and $\lambda_C$ denotes the weight of $\mathcal{R}_C$. It is noteworthy that the Data Simulator component is designed to be modular, allowing for instantiation or replacement with any generative model capable of incorporating prior knowledge.

## 3.4 ANALYSIS ON THE PRIOR KNOWLEDGE IN DATA SIMULATOR

In the data simulator, the estimated correlation matrix $\hat{\mathcal{C}}_{T+1}$ serves as prior knowledge and enhances the quality of concept drift data generation. Notice that the high-quality concept drift data generation is critical to guarantee the well-performed model in TDG, the analysis of the prior knowledge (i.e., low feature correlation matrix distance) in Eq. (5). In this subsection, we provide Theorem 1 to investigate the rationale of this prior knowledge. The proof of Theorem 1 is provided in Appendix A.

**Theorem 1.** *Considering two $d$-dimensional variables $\mathbf{U} = (U_1, U_2, \cdots, U_d)$ and $\mathbf{V} = (V_1, V_2, \cdots, V_d)$, assume (i) bounded variable: $|U_i| \leq A$ and $|V_i| \leq A$ for any $i = 1, 2, \cdots, d$, where $A$ is the maximum value of variable; (ii) positive variance: $\min\{\mathbb{D}[U_i], \mathbb{D}[V_i]\} \geq \delta$, where $\mathbb{D}[\cdot]$ represents the variance of random variable, and $\delta$ is the lower bound of variance; (iii) Similar distribution of multi-variables: $TV(\mathbf{U}, \mathbf{V}) \leq \epsilon$, where $TV(\cdot, \cdot)$ represents total variation distance, and $\epsilon$ is the distribution distance tolerance. Under assumptions (i), (ii), and (iii), the relation between the feature correlation matrices $\mathcal{C}_{\mathbf{U}}$ and $\mathcal{C}_{\mathbf{V}}$ is given by*

$$\|\mathcal{C}_{\mathbf{U}} - \mathcal{C}_{\mathbf{V}}\|_1 \leq 6d\epsilon A^2 \left(\frac{1}{\delta} + \frac{A^2}{\delta^2}\right). \tag{6}$$

**Discussion on assumptions (i), (ii), (iii).** We clarify that these assumptions can be easily satisfied in the real world. The assumption (i) of a bounded variable is naturally satisfied and can be easily reshaped by feature normalization pre-processing. The assumption (ii) of positive variance is also naturally satisfied by pre-processing. The constant variable (feature) represents there is not any information for such a feature and should be filtered out. As the assumption (iii), the data distribution shift is usually smoothed and the distribution of data collected within a short time period should be similar.

**Remarks.** Theorem 1 demonstrates that the prior knowledge in Eq. (5) can be guaranteed under assumptions (i), (ii), (iii), which serves as the theoretical foundation for the usage of prior knowledge in data simulator. The derived bound is tight when the data shift is zero, i.e., $\epsilon = 0$. Furthermore, our analysis suggests that prior knowledge, such as a correlation matrix, may not effectively represent high-dimensional data, and generating such high-dimensional data poses a significant challenge.

## 4 EXPERIMENT

In this section, we conduct experiments to evaluate the performance of CODA, aiming to answer the following three research questions: **RQ1:** How effective is the simulated future data by CODA to train different model architectures for temporal domain generalization? **RQ2:** How is the quality of the simulated future data by CODA? **RQ3:** Does the predicted feature correlation contribute to generating concept drift future data?

### 4.1 EXPERIMENT SETTING

**Datasets.** To evaluate the efficacy of our proposed model-agnostic framework, CODA, we conduct experiments on four classification datasets—Rotated Moons (2-Moons), Electrical Demand (Elec2), Online News Popularity (ONP), and Shuttle—as well as one regression dataset, Appliances Energy Prediction (Appliance). While the 2-Moons is a synthetic dataset with rotation angle acting as a temporal proxy, the remaining four datasets are real-world, temporally evolving collections. More details about the datasets can be found in Appendix C.

**Baseline Methods.** We compare our data-centric approach with three groups of methods: **Time-Oblivious Baselines**: These methods disregard concept drift, including the strategies Offline (trained on all source domains) and LastDomain (trained on the last source domain); **Continuous Domain Adaptation**: These algorithms focus on transporting either the last or historical source domains to future domains, including CDOT (Ortiz-Jimenez et al., 2019), CIDA (Wang et al., 2020), and Adagraph (Mancini et al., 2019); and **Temporal Domain Generalization**: These approaches specifically tackle temporal distribution shifts, including GI (Nasery et al., 2021) and DRAIN (Bai et al., 2022). More baseline details can be found in Appendix D.

**Implementation Details.** In the CODA framework, we employ LSTM (Hochreiter & Schmidhuber, 1997) for the Correlation Predictor and instantiate the Data Simulator component with GOGGLE (Liu et al., 2022). To enhance the compatibility of prior knowledge with the Data Simulator, we extract Feature Correlation matrices from each source domain dataset using GOGGLE's learnable relational structure. All the experiments of CODA are repeated five times on each dataset, with the average results and standard deviation in Table 1. To evaluate the capability of CODA in generating concept-drift data for model training, we implement three predictive architectures: Multilayer Perceptron (MLP), the tree-based LightGBM (Ke et al., 2017), and the Transformer-based FT-Transformer (Gorishniy et al., 2021). More implementation details are elaborated in Appendix D.

### 4.2 QUANTITATIVE ANALYSIS (RQ1)

To assess the generalization efficacy of CODA in the presence of concept drift across temporal domains, we employ CODA to generate future data, matching the sample size of the unseen domains across five datasets. Subsequently, we compare the performance of models trained on this generated data with other baseline methods. To further assess the advantage of our model-agnostic approach, we utilize three different model architectures for the evaluation. For the Data Simulator component's model selection, we employ MLPs with hyperparameters aligned to those of the baseline methods. Subsequently, we fine-tune the hyperparameters of LightGBM and FT-Transformer based on the

Table 1: Performance comparison: Misclassification error rates (in %) for classification tasks and Mean Absolute Error (MAE) for regression tasks, both are lower the better. Results of all methods, excluding "Shuttle," are reported from Bai et al. (2022).

| | Classification (in %) | | | | Regression |
|---|---|---|---|---|---|
| | 2-Moons | Elec2 | ONP | Shuttle | Appliance |
| Offline | $22.4 \pm 4.6$ | $23.0 \pm 3.1$ | $\mathbf{33.8 \pm 0.6}$ | $7.2 \pm 0.1$ | $10.2 \pm 1.1$ |
| LastDomain | $14.9 \pm 0.9$ | $25.8 \pm 0.6$ | $36.0 \pm 0.2$ | $6.7 \pm 0.0$ | $9.1 \pm 0.7$ |
| IncFinetune | $16.7 \pm 3.4$ | $27.3 \pm 4.2$ | $34.0 \pm 0.3$ | $7.0 \pm 0.1$ | $8.9 \pm 0.5$ |
| CDOT (Ortiz-Jimenez et al., 2019) | $9.3 \pm 1.0$ | $17.8 \pm 0.6$ | $34.1 \pm 0.0$ | $6.5 \pm 0.2$ | - |
| CIDA (Wang et al., 2020) | $10.8 \pm 1.6$ | $14.1 \pm 0.2$ | $34.7 \pm 0.6$ | - | $8.7 \pm 0.2$ |
| Adagraph (Mancini et al., 2019) | $8.0 \pm 1.1$ | $20.1 \pm 2.2$ | $40.9 \pm 0.6$ | $6.9 \pm 0.2$ | - |
| GI (Nasery et al., 2021) | $3.5 \pm 1.4$ | $16.9 \pm 0.7$ | $36.4 \pm 0.8$ | $7.0 \pm 0.1$ | $8.2 \pm 0.6$ |
| DRAIN (Bai et al., 2022) | $3.2 \pm 1.2$ | $12.7 \pm 0.8$ | $38.3 \pm 1.2$ | $7.4 \pm 0.3$ | $6.4 \pm 0.4$ |
| CODA (MLP) | $\mathbf{2.3 \pm 1.0}$ | $\mathbf{10.3 \pm 1.1}$ | $36.5 \pm 0.4$ | $\mathbf{6.3 \pm 0.1}$ | $\mathbf{4.6 \pm 0.5}$ |
| CODA (LightGBM) | $\mathbf{1.4 \pm 0.4}$ | $\mathbf{10.6 \pm 0.6}$ | $41.8 \pm 0.3$ | $\mathbf{6.1 \pm 0.2}$ | $\mathbf{5.3 \pm 0.2}$ |
| CODA (FT-Transformer) | $\mathbf{0.5 \pm 0.2}$ | $\mathbf{11.0 \pm 0.4}$ | $42.1 \pm 0.4$ | $6.6 \pm 0.1$ | $\mathbf{5.3 \pm 0.3}$ |

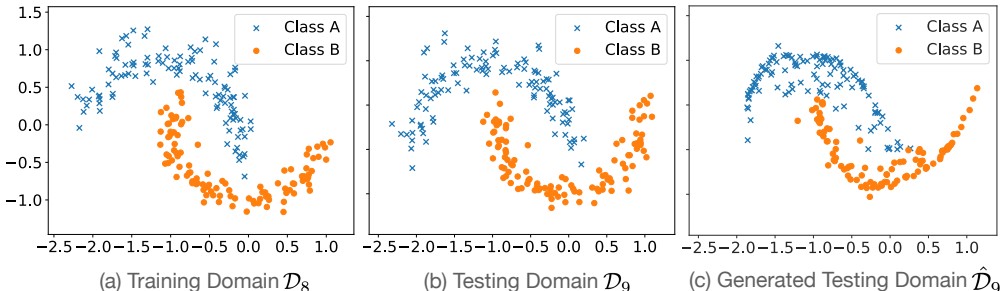

(a) Training Domain $\mathcal{D}_8$     (b) Testing Domain $\mathcal{D}_9$     (c) Generated Testing Domain $\hat{\mathcal{D}}_9$

Figure 3: Visualization of generated future domain data in 2-Moons dataset, where $\mathcal{D}_8$ is the last source domain, $\mathcal{D}_9$ is the unseen test domain, and $\hat{\mathcal{D}}_9$ is the synthetic domain generated by CODA.

validation set within the generated future data. The comparison results are summarized in Table 1. Overall, the three architectures trained on data generated by CODA achieve the best Misclassification Error rates (McE) in three of the four classification datasets and the best Mean Absolute Error (MAE) in the regression dataset. The only exception is the ONP dataset, where the Offline method outperforms all other approaches. This is consistent with previous research indicating that the ONP dataset exhibits relatively weak concept drift (Nasery et al., 2021).

Moreover, the results demonstrate that the future data generated by CODA is amenable to training across different model architectures. It also demonstrates that the optimal architecture might intrinsically depend on the specific data modality. As shown in Table 1, distinct model architectures excel in different datasets. For instance, LightGBM and FT-Transformer surpass MLP in performance on the 2-Moons dataset, yet fail to achieve the lowest McE on the Elec2 and Appliance datasets.

**Observation 1: CODA is free from fixed in specific model architectures by providing temporal generalization data for training.** CODA provides a model-agnostic approach to simulate future domain data, tackling concept drift at its fundamental cause. Its data-centric manner allows different architectures to be fine-tuned on the simulated data for achieving temporal domain generalization.

### 4.3 QUALITY OF GENERATED FUTURE DATA (RQ2)

**Visualization of the generated future data.** As mentioned in Section 3, CODA aims to provide high-fidelity simulated data. To illustrate the quality of the predicted future data, we compare the distributions of the generated domain data $\hat{\mathcal{D}}_9$ with the unseen domain $\mathcal{D}_9$ in the 2-Moon dataset. Recall that the 2-Moons dataset is synthesized with each domain $i$ characterized by a rotation angle of $18i^\circ$. Consequently, the generated $\hat{\mathcal{D}}_9$ should be rotated by the same angle as $\mathcal{D}_9$. As shown in Figure 3, $\hat{\mathcal{D}}_9$ closely resembles $\mathcal{D}_9$, and both can be distinguished from the last source domain $\mathcal{D}_8$ by a specific rotation angle. More visualization analysis are available in Appendix E.

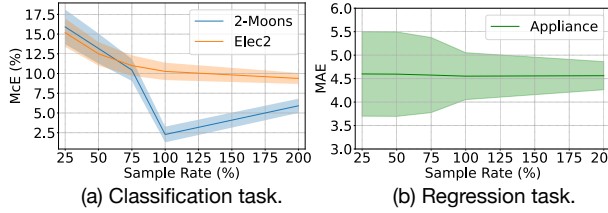

(a) Classification task.  (b) Regression task.

Figure 4: Sample rate variations in CODA: Evaluating McE and MAE using MLPs trained on generated samples, where the sample rate is the percentage of the number of testing domain samples.

Table 2: Ablation study on the exclusion of the feature correlation matrix $\hat{\mathcal{C}}_{T+1}$. Here, *w/o* $\hat{\mathcal{C}}_{T+1}$ refers a variant of CODA that does not incorporate $\hat{\mathcal{C}}_{T+1}$ as prior knowledge.

| | Classification (in %) | | Regression |
|---|---|---|---|
| | 2-Moons | Elec2 | Appliance |
| CODA | $\mathbf{2.3 \pm 1.0}$ | $\mathbf{10.3 \pm 1.1}$ | $\mathbf{4.6 \pm 0.5}$ |
| *w/o* $\hat{\mathcal{C}}_{T+1}$ | $15.6 \pm 0.7$ | $26.5 \pm 0.8$ | $9.5 \pm 0.4$ |

**Impact of numbers of the generated future data.** As the CODA provides a framework for training a temporal domain generalization generative model, we can decide the number of generated samples for model training. Thus, we can investigate the influence of the number of the generated samples on the trained prediction model, here we use MLP as the prediction model architecture. CODA provides a framework for training a generative model to achieve temporal domain generalization. This data-centric design grants us the ability to modulate the number of generated samples for model training. Here, we investigate the impacts of varying sample counts on the performance of an MLP trained with the generated data. As shown in Figure 4, for both classification and regression datasets, increasing the sample rate reduces performance variances because a larger dataset more accurately represents the probability distribution learned by the Data Simulator. Regarding performance, in the real-world classification dataset, Elec2, increasing the number of generated samples enhances the trained model's performance. However, in the low-dimensional toy dataset, 2-Moons, generating additional data might introduce noise, potentially degrading the performance. This might indicate the representative of the moderate quantity of the generated samples. On the other hand, in the real-world regression dataset, Appliance, the generated data with 25% sample rate is representative enough for achieving stable model performance with marginally higher variance, which also indicates the quality of the samples simulated by CODA.

**Observation 2: CODA is able to simulate high-quality samples for training temporal domain generalization models.** Visualizations and sample rate analyses demonstrate that CODA effectively generates high-quality future domain data for training, and these generated samples offer a representative data distribution in the presence of concept drift.

**Cross-Architecture Transferability.** Since the outcome of CODA is a dataset for model training, it can conceptually be employed by any model architecture. Thus, once the hyperparameters of the Data Simulator are tuned for one architecture, the generated data can readily be utilized to train other predictive models. To examine such transferability, we conduct experiments where we tune the Data Simulator using three distinct architectures: MLP, LightGBM, and FT-Transformer (FTT). Subsequently, we train the other two architectures on the dataset generated by CODA. The results indicate the generated data is transferable. As shown in Table 3, across the Data Simulator tuned by three different architectures, LightGBM achieves the best average performance. Interestingly, LightGBM outperforms FTT even when both are trained on the data generated by the FTT-tuned Data Simulator, suggesting the advantage of this data-centric approach.

**Observation 3: CODA-generated future data is transferable to different model architectures.** The transferability analysis demonstrates that CODA-generated data can be used as training data for MLP-based, tree-based, and transformer-based models. Additionally, due to the less hyperparameter search space, we suggest utilizing MLP-based models for tuning the Data Simulator and then training different prediction model architectures on the generated data.

## 4.4 Ablation Study on Prior Knowledge Matrix (RQ3)

To further elucidate the impact of the predicted feature correlation matrix $\hat{\mathcal{C}}_{T+1}$, we conduct ablation studies on observing the effectiveness of prior knowledge in CODA. Notably, *w/o* $\hat{\mathcal{C}}_{T+1}$ refers to a variant of CODA that excludes $\hat{\mathcal{C}}_{T+1}$ during training the Data Simulator component. According to

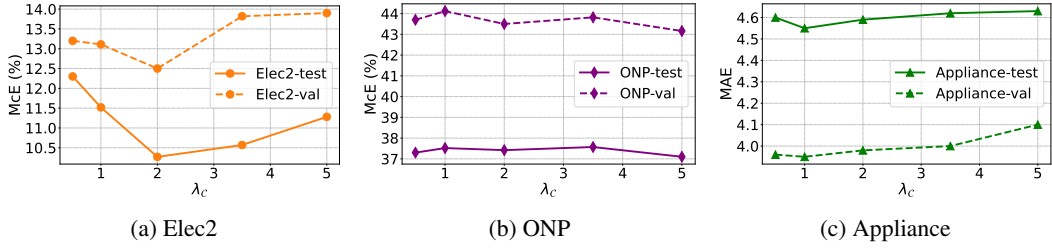

Figure 5: Sensitivity analysis on the $\lambda_{\mathcal{C}}$, where the trained model architecture is MLP.

Equation 4 and 5, *w/o* $\hat{\mathcal{C}}_{T+1}$ essentially trains the Data Simulator only using the last source domain. In Table 2, for every dataset, all results are conducted by using the same MLP architecture, albeit trained on different data. The results suggest that the data generated by *w/o* $\hat{\mathcal{C}}_{T+1}$ closely mirrors the characteristics of the last source domain when used as training data, as the LastDomain in Table 1. Conversely, CODA is capable of providing a training set that captures the essence of concept drift by incorporating the feature correlation $\hat{\mathcal{C}}_{T+1}$ during the training of the Data Simulator. As a result, the identical architecture, when trained on the dataset simulated by CODA, demonstrates markedly superior temporal domain generalization. More analysis of the predicted feature correlation matrices is available in Appendix F.

## 4.5 SENSITIVITY ANALYSIS (RQ3)

To assess the significance of the predicted correlation matrix of CODA framework, we analyze the sensitivity impact of the weighted hyperparameter $\lambda_{\mathcal{C}}$ as presented in Equation 5 with the MLP architecture. As shown in Figure 5, despite the optimal $\lambda_{\mathcal{C}}$ varies among different datasets, the trend of using $\lambda_{\mathcal{C}}$ can be identified on validation sets, where the trend of obtaining best hyperparameter settings in validation sets is similar to the one in test sets. In the two real-world datasets, Elec2 and Appliance, the performance trend with respect to $\lambda_{\mathcal{C}}$ remains consistent between the validation and future testing sets. Conversely, in the ONP dataset, proven to exhibit almost no concept drift, there is no specific performance trend. This might suggest no significant concept drift that can be captured by CODA.

Table 3: Transferability. Note that "Tune" represents models used for searching hyperparameter of the Data Simulator, while "Train" refers to models trained on the CODA-generated data.

| Tune\Train | MLP | LightGBM | FTT |
|---|---|---|---|
| MLP | **10.3 ± 1.1** | 10.6 ± 0.4 | 11.0 ± 0.4 |
| LightGBM | 10.6 ± 0.9 | **9.1 ± 0.3** | 12.2 ± 0.5 |
| FTT | 11.6 ± 1.1 | **10.3 ± 0.4** | 10.6 ± 0.4 |
| Avg. | 10.8 ± 1.0 | **10.0 ± 0.4** | 11.3 ± 0.4 |

**Observation 4: The dataset-sensitive hyperparameter $\lambda_{\mathcal{C}}$ is identifiable on the concept drift datasets.** The sensitivity analysis reveals that, in the presence of concept drift, the performance trend tied to $\lambda_{\mathcal{C}}$ remains consistent between the validation set and the unseen future domain, underscoring the feasibility of CODA in real-world settings.

## 5 CONCLUSION AND FUTURE WORK

In this work, we demonstrate the efficacy of tackling the fundamental cause of concept drift from a data perspective. To address the limited model prediction ability to capture the underlying temporal trends among the chronological data distributions of source domains, we propose a two-step data simulation framework, CODA, incorporating feature correlation matrices to capture temporal trends within the historical source domains. Specifically, CODA extracts the temporal evolving of historical feature correlation matrices to predict the correlation matrix for the upcoming future domain, and then simulates future training data based on this predicted correlation. Theoretical analysis guarantees that the predicted correlation is reliable under practical assumptions. Experimental results demonstrate that the CODA-generated data can be used as the training data for different model architectures to achieve temporal domain generalization. Regarding future directions, CODA framework unveils several potential facets that merit further exploration, such as more accurate future correlation prediction and advanced generative models to utilize predicted prior knowledge.

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

APPENDIX

## A    PROOF OF THEOREM 1

Before delving into the proof, we would like to introduce some primary knowledge on total variation distance and its dual-representative format.

**Definition 1** (Total variation distance). *Let $\mathcal{P}_U$, $\mathcal{P}_V$ be two probability distributions for random variables $U$ and $V$ over a shared probability space $\Omega$. Then, the total variation distance between them, denoted $TV(U, V)$, is given by*

$$TV(U, V) = \sup_{\mathcal{E} \subseteq \Omega} |\mathcal{P}_U(\mathcal{E}) - \mathcal{P}_V(\mathcal{E})|. \tag{7}$$

**Lemma 1** (Dual representation of TV distance). *Let $\mathcal{P}_U$, $\mathcal{P}_V$ be two probability distributions for random variables $U$ and $V$ over a shared probability space $\Omega$, the dual representation of total variation distance is given by*

$$TV(U, V) = \frac{1}{2K} \sup_{\|f\|_\infty \leq K} \left| \mathbb{E}_{\mathcal{P}_U}[f(x)] - \mathbb{E}_{\mathcal{P}_V}[f(x)] \right|, \tag{8}$$

*where $l_\infty$ norm for function $\|f\|_\infty$ represents the maximum value of function, i.e., $\|f\|_\infty = \inf\{C > 0 : |f(x)| \leq C$ for almost everywhere$\}$, and $K$ could be any constant.*

Recall that there are three assumptions (i) bounded variable: $|U_i| \leq A$ and $|V_i| \leq A$ for any $i = 1, 2, \cdots, d$; (ii) positive variance: $\min\{\mathbb{D}[U_i], \mathbb{D}[V_i]\} \geq \delta$; (iii) Similar distribution of multi-variables: $TV(\mathbf{U}, \mathbf{V}) \leq \epsilon$, we aim to prove the upper bound for the feature correlation matrices $\mathcal{C}_{\mathbf{U}}$ and $\mathcal{C}_{\mathbf{V}}$. In the following part, we use the notation for brevity: $\Delta E_i \triangleq |\mathbb{E}[U_i] - \mathbb{E}[V_i]|$, $\Delta E_{ij} \triangleq |\mathbb{E}[U_i U_j] - \mathbb{E}[V_i V_j]|$, and $\Delta D_i \triangleq |\mathbb{D}[U_i] - \mathbb{D}[V_i]|$, where $\mathbb{E}[\cdot]$ and $\mathbb{D}[\cdot]$ represents the expectation and variance of random variable, respectively.

Firstly, we can provide the bounds of the expectation and variance difference as follows:

**Lemma 2.** *Under assumptions (i) and (iii), we have the following bounds on the expectation and variance difference:*

$$\Delta E_i \triangleq |\mathbb{E}[U_i] - \mathbb{E}[V_i]| \leq 2\epsilon A, \tag{9}$$

$$\Delta E_{ij} \triangleq |\mathbb{E}[U_i U_j] - \mathbb{E}[V_i V_j]| \leq 2\epsilon A^2, \tag{10}$$

$$\Delta D_i \triangleq |\mathbb{D}[U_i] - \mathbb{D}[V_i]| \leq 6\epsilon A^2. \tag{11}$$

*Proof.* For the bound on the expectation difference, based on Lemma 1 with $f_0(x) = x$, it is easy to obtain

$$\Delta E_i \triangleq |\mathbb{E}[U_i] - \mathbb{E}[V_i]| \overset{(a)}{\leq} 2TV(U_i, V_i)\|f_0\|_\infty \leq 2TV(\mathbf{U}, \mathbf{V})\|f_0\|_\infty \leq 2\epsilon A; \tag{12}$$

where inequality (a) is based on dual representation of TV distance (Lemma 1). For the bound on the covariance difference, based on Lemma 1 with $f_1(\mathbf{x}) = x_i x_j$, it is easy to obtain

$$\Delta E_{ij} \triangleq |\mathbb{E}[U_i U_j] - \mathbb{E}[V_i V_j]| \leq 2TV(\mathbf{U}, \mathbf{V})\|f_1\|_\infty \leq 2\epsilon A^2; \tag{13}$$

Similarly, we can also obtain $|\mathbb{E}[U_i^2] - \mathbb{E}[V_i^2]| \leq \epsilon A^2$. Therefore, the bound on the variance difference is given by

$$\Delta D_i \triangleq |\mathbb{D}[U_i] - \mathbb{D}[V_i]| \leq |\mathbb{E}[U_i^2] - \mathbb{E}[V_i^2]| + |\mathbb{E}[U_i] - \mathbb{E}[V_i]||\mathbb{E}[U_i] + \mathbb{E}[V_i]|,$$
$$\leq 2\epsilon A^2 + 2\epsilon A \cdot 2A = 6\epsilon A^2. \tag{14}$$

$\square$

*Subsequently, we consider the correlation coefficient distance on the $i$-row $j$-column element $|C_{\mathcal{U},ij} - C_{\mathcal{U},ij}|$, where $C_{\mathcal{U},ij} = \frac{\mathbb{E}[U_iU_j]-\mathbb{E}[U_i]\mathbb{E}[U_j]}{\sqrt{\mathbb{D}[U_i]\cdot\mathbb{D}[U_i]}}$. According to basic algebra, we have*

$$
\begin{aligned}
|C_{\mathcal{U},ij} - C_{\mathcal{U},ij}| &= \left| \frac{\mathbb{E}[U_iU_j] - \mathbb{E}[U_i]\mathbb{E}[U_j]}{\sqrt{\mathbb{D}[U_i]\cdot\mathbb{D}[U_j]}} - \frac{\mathbb{E}[V_iV_j] - \mathbb{E}[V_i]\mathbb{E}[V_j]}{\sqrt{\mathbb{D}[V_i]\cdot\mathbb{D}[V_j]}} \right|, \\
&= \left| \frac{\mathbb{E}[U_iU_j] - \mathbb{E}[U_i]\mathbb{E}[U_j]}{\sqrt{\mathbb{D}[U_i]\cdot\mathbb{D}[U_j]}} - \frac{\mathbb{E}[V_iV_j] - \mathbb{E}[V_i]\mathbb{E}[V_j]}{\sqrt{\mathbb{D}[U_i]\cdot\mathbb{D}[U_j]}} \right. \\
&\quad + \left. \frac{\mathbb{E}[V_iV_j] - \mathbb{E}[V_i]\mathbb{E}[V_j]}{\sqrt{\mathbb{D}[U_i]\cdot\mathbb{D}[U_j]}} - \frac{\mathbb{E}[V_iV_j] - \mathbb{E}[V_i]\mathbb{E}[V_j]}{\sqrt{\mathbb{D}[V_i]\cdot\mathbb{D}[V_j]}} \right| \\
&\leq \underbrace{\frac{\left| \mathbb{E}[U_iU_j] - \mathbb{E}[V_iV_j] + \mathbb{E}[V_i]\mathbb{E}[V_j] - \mathbb{E}[U_i]\mathbb{E}[U_j] \right|}{\sqrt{\mathbb{D}[U_i]\cdot\mathbb{D}[U_j]}}}_{\mathcal{I}_1} \\
&\quad + \underbrace{\left| \mathbb{E}[V_iV_j] - \mathbb{E}[V_i]\mathbb{E}[V_j] \right| \left| \frac{1}{\sqrt{\mathbb{D}[U_i]\cdot\mathbb{D}[U_j]}} - \frac{1}{\sqrt{\mathbb{D}[V_i]\cdot\mathbb{D}[V_j]}} \right|}_{\mathcal{I}_2}
\end{aligned} \tag{15}
$$

*For the first term $\mathcal{I}_1$, we can obtain*

$$
\begin{aligned}
\mathcal{I}_1 &\leq \frac{1}{\sqrt{\mathbb{D}[U_i]\cdot\mathbb{D}[U_j]}} \left( \left| \mathbb{E}[U_iU_j] - \mathbb{E}[V_iV_j] \right| + \left| \mathbb{E}[V_i]\mathbb{E}[V_j] - \mathbb{E}[U_i]\mathbb{E}[U_j] \right| \right), \\
&\leq \frac{1}{\sqrt{\mathbb{D}[U_i]\cdot\mathbb{D}[U_j]}} \left( \left| \mathbb{E}[U_iU_j] - \mathbb{E}[V_iV_j] \right| + \mathbb{E}[V_i]\left| \mathbb{E}[U_j] - \mathbb{E}[V_j] \right| + \mathbb{E}[V_j]\left| \mathbb{E}[U_i] - \mathbb{E}[V_i] \right| \right) \\
&\leq \frac{1}{\delta}(\epsilon A^2 + \epsilon A^2 + \epsilon A^2) = \frac{1}{\delta}3\epsilon A^2.
\end{aligned} \tag{16}
$$

*For the second term $\mathcal{I}_2$, we can also obtain*

$$
\begin{aligned}
\mathcal{I}_1 &\leq A^2 \left| \frac{1}{\sqrt{\mathbb{D}[U_i]\cdot\mathbb{D}[U_j]}} - \frac{1}{\sqrt{\mathbb{D}[V_i]\cdot\mathbb{D}[V_j]}} \right| \\
&\leq A^2 \left| \frac{1}{\sqrt{\mathbb{D}[U_i]\cdot\mathbb{D}[U_j]}} - \frac{1}{\sqrt{\mathbb{D}[V_i]\cdot\mathbb{D}[U_j]}} + \frac{1}{\sqrt{\mathbb{D}[V_i]\cdot\mathbb{D}[U_j]}} - \frac{1}{\sqrt{\mathbb{D}[V_i]\cdot\mathbb{D}[V_j]}} \right| \\
&\leq A^2 \left( \frac{\mathbb{D}[U_j] - \mathbb{D}[V_j]}{\sqrt{\mathbb{D}[U_i]\mathbb{D}[U_j]\mathbb{D}[V_j]}(\sqrt{\mathbb{D}[U_j]} + \sqrt{\mathbb{D}[V_j]})} + \frac{\mathbb{D}[U_i] - \mathbb{D}[V_i]}{\sqrt{\mathbb{D}[V_j]\mathbb{D}[U_i]\mathbb{D}[V_i]}(\sqrt{\mathbb{D}[U_i]} + \sqrt{\mathbb{D}[V_i]})} \right) \\
&\leq A^2 \left( \frac{6\epsilon A^2}{2\delta^2} + \frac{6\epsilon A^2}{2\delta^2} \right) = \frac{6\epsilon A^4}{\delta^2}.
\end{aligned} \tag{17}
$$

*Therefore, we have the correlation coefficient distance on the $i$-row $j$-column element as follows:*

$$
|C_{\mathcal{U},ij} - C_{\mathcal{U},ij}| = 6\epsilon A^2 \left( \frac{1}{\delta} + \frac{A^2}{\delta^2} \right). \tag{18}
$$

*Finally, the relation between the feature correlation matrices can be given by*

$$
\|\mathcal{C}_{\mathbf{U}} - \mathcal{C}_{\mathbf{U}}\|_1 = \max_{1 \leq k \leq d} \sum_{i=1}^{d} |C_{\mathcal{U},ij} - C_{\mathcal{U},ij}| \leq 6d\epsilon A^2 \left( \frac{1}{\delta} + \frac{A^2}{\delta^2} \right). \tag{19}
$$

# B ALGORITHM OF CODA

## B.1 ALGORITHM OF CODA

The training outline of CODA is given in Algorithm 1. CODA follows equation 3 for capturing temporal trends of feature correlation (line 5-6). After $\boldsymbol{H}(\cdot)$ is converged for $\hat{\mathcal{C}}_{T+1}$, the Data Simulator $\boldsymbol{G}(\mathcal{D}_T; \hat{\mathcal{C}}_{T+1}|\theta_G)$ can be updated by Equation 5 and terminated when it is converged. Overall, $\boldsymbol{G}(\mathcal{D}_T; \hat{\mathcal{C}}_{T+1}|\theta_G)$ learns to approximate both the observed data distribution $\mathcal{D}_T$ and the predicted temporal feature correlation $\hat{\mathcal{C}}_{T+1}$ for simulating the future data under concept drift.

---

**Algorithm 1** Two-stage Future Data Generation Training with CODA

---

1: **Input:**
   Feature Correlation Matrices of each source domain time points $\mathcal{C}_1, \mathcal{C}_2, \ldots, \mathcal{C}_T$
   Correlation Predictor (an RNN-based model) $\boldsymbol{H}(\cdot)$
   Current source domain data $\mathcal{D}_T$
   Data Simulator (a generative model) $\boldsymbol{G}(\mathcal{D}_i; \mathcal{C}_{i+1}|\theta_G)$
2: **Output:**
   Temporal domain generalization data simulator $\boldsymbol{G}(\mathcal{D}_T; \hat{\mathcal{C}}_{T+1}|\theta_G)$
3: **First stage: Correlation Predictor Training**
4: **while** not convergence **do**
5:     Predict the feature correlation matrix of the next domain data by $\boldsymbol{H}(\cdot)$ and Equation 2.
6:     Update $\hat{\mathcal{C}}_{T+1} = \boldsymbol{H}(\mathcal{C}_1, \mathcal{C}_2, \ldots, \mathcal{C}_T)$ by Equation 3.
7: **end while**
8: **Second stage: Data Simulator Training**
9: **while** not convergence **do**
10:     Generate future data distribution with $\hat{\mathcal{C}}_{T+1}$ and Data Simulator $\hat{\mathcal{D}}_{T+1} = \boldsymbol{G}(\mathcal{D}_{\mathcal{T}}; \hat{\mathcal{C}}_{T+1}|\theta_G)$.
11:     Update $\boldsymbol{G}(\mathcal{D}_T; \hat{\mathcal{C}}_{T+1}|\theta_G)$ by Equation 5.
12: **end while**

---

# C DATASETS DETAILS

In this section, we provide a detailed description of the five datasets employed in our experiments.

**Rotated 2 Moons (2-Moons).** This dataset is a modified version of the 2-entangled moons, where the lower moon is labeled as 0 and the upper moon as 1. Each moon comprises 100 instances. We generate 10 domains by sampling 200 data points from the 2-Moons distribution and rotating them counter-clockwise in increments of $18°$. Consequently, domain $i$ undergoes a rotation of $18i$. Domains 0 through 8 serve as our training domains, while domain 9 is reserved for testing.

**Electrical Demand (Elec2).** This dataset captures the electricity demand in a specific province. It comprises 8 features, including price, day of the week, and units transferred. The binary classification task involves predicting whether the electricity demand in each 30-minute interval was above or below the average demand of the previous day. Instances with missing values are excluded. We define two weeks as one time domain. The model is trained on 29 domains and tested on the 30th domain, resulting in 27,549 training points and 673 test points.

Table 4: Notations in this paper.

| Notation | Description |
|---|---|
| $\mathcal{D}_t$ | Source domain dataset at time $t$ |
| $\hat{\mathcal{D}}_t$ | Predicted domain dataset at time $t$ |
| $N_t$ | Sample size at time $t$ |
| $\mathcal{X}_t, \mathcal{Y}_t$ | Feature / label space at time $t$ |
| $x^t, y^t$ | Sample / label at time $t$ |
| $\mathcal{C}_t$ | Feature correlation matrix at time $t$ |
| $\hat{\mathcal{C}}_t$ | Predicted correlation matrix at time $t$ |
| $\boldsymbol{H}(\cdot)$ | Correlation Predictor |
| $\boldsymbol{G}(\cdot; \cdot|\theta_G)$ | Data Simulator |
| $\theta_G$ | Parameters of $\boldsymbol{G}$ |
| $\mathcal{P}(\cdot)$ | Probability Distribution |

**Online News Popularity (ONP).** Originating from Fernandes et al. (2015), this dataset aggregates a diverse set of features about articles published by Mashable over a two-year span. The objective is to predict the article's popularity based on the number of shares in social networks. The dataset

is temporally split into six domains, with the initial five designated for training. While the concept drift is manifested through the progression of time, prior studies, such as Nasery et al. (2021), have demonstrated that the drift is relatively mild.

**Shuttle**[3]**.** The Shuttle dataset comprises approximately 58,000 data points, each with 9 features, pertaining to space shuttles in flight. The objective is multi-class classification, characterized by a pronounced class imbalance. Data points are temporally divided into eight domains based on their associated timestamps. Specifically, timestamps ranging from 30 to 70 define the training domains, while those from 70 to 80 delineate the test domain.

**Appliances Energy Prediction (Appliance).** This dataset Candanedo et al. (2017) is used to create regression models of appliances energy use in a low energy building. The data set is at 10 min for about 4.5 months in 2016, and we treat each half month as a single domain, resulting in 9 domains in total. The first 8 domains are used for training and the last one is for testing. Similar to Elec2, the drift for this dataset corresponds to how the appliances energy usage changes in a low-energy building over about 4.5 months in 2016.

# D  BASELINES AND IMPLEMENTATION DETAILS

## D.1  BASELINES.

**Practical Baseline.** (i) Offline: A time-oblivious model trained using Empirical Risk Minimization (ERM) across all source domains. (ii) LastDomain: A time-oblivious model trained using ERM exclusively on the most recent source domain. (iii) IncFinetune: This approach biases the training towards more recent data. Initially, the Baseline method is applied to the first time point. Subsequently, the model is fine-tuned on successive time points in a sequential manner, using a reduced learning rate. In this paper, we follow the settings from Nasery et al. (2021) and Bai et al. (2022) for the implementations of the aforementioned three practical baselines.

**Continuous Domain Adaptation Methods.** (i) CDOT (Ortiz-Jimenez et al., 2019): This model transports the most recent labeled examples $D^T$ to the future using a learned coupling from past data, and trains a classifier on them. (ii) CIDA (Wang et al., 2020): This method is representative of typical domain erasure methods applied to continuous domain adaptation problems. (iii) Adagraph (Mancini et al., 2019): This approach renders the batch normalization parameters time-sensitive and applies a specified kernel for smoothing.

**Temporal Domain Generalization Method.** (i) GI (Nasery et al., 2021): This approach introduces a training algorithm that encourages models to learn functions capable of extrapolating effectively to the near future. It achieves this by supervising the first-order Taylor expansion of the learned function. (ii) DRAIN (Bai et al., 2022): This method presents a framework that predicts MLP weights for predicting a near-future domain. It does so by leveraging an LSTM unit and considering the context of chronological source domain MLP weights.

## D.2  IMPLEMENTATIONS.

For all the experiments in this paper, we instantiate the two components of CODA, Correlation Predictor 3.2 and Data Simulator 3.3, with LSTM (Hochreiter & Schmidhuber, 1997) and GOGGLE (Liu et al., 2022). We use Adam optimizer for all the experiments. For hyperparameter tuning of the Correlation Predictor, we consider the following search ranges: learning rate: $1 \times 10^{-5}$ to $1 \times 10^{-2}$; number of LSTM layers: 4 to 16; LSTM latent dimension: 4 to 16; LSTM hidden dimension: 4 to 16; $\lambda_{CE}$ in Equation 3: 0 to 20. For hyperparameter tuning of the Data Simulator, we consider the following search ranges: learning rate: $1 \times 10^{-5}$ to $1 \times 10^{-2}$; encoder dimension: 48 to 72; encoder layer: 2 to 4; decoder dimension: 48 to 72; decoder layer: 2 to 4; $\lambda_C$ in Equation 5: 0.1 to 20.

For each dataset, the specific hyperparameters associated with the CODA components are detailed in Table 5. The hyperparameters for the architectures trained on the CODA-generated data (MLP,

---

[3]https://archive.ics.uci.edu/dataset/148/statlog+shuttle

Table 5: Hyperparameters of CODA. Here, "L. dim" denotes the latent dimension, "H. dim" represents the hidden dimension, $\lambda_{CE}$ is the weight from Equation 3, "En. dim" indicates the encoder dimension, "En. layer" specifies the number of encoder layers, "De. dim" refers to the decoder dimensions, "De. layer" signifies the number of decoder layers, and $\lambda_{\mathcal{C}}$ is the weight from Equation 5.

| | Correlation Predictor | | | | | Data Simulator | | | | |
|---|---|---|---|---|---|---|---|---|---|---|
| | Lr | Layer | L. dim | H. dim | $\lambda_{CE}$ | Lr | En. dim | En. layer | De. dim | De. layer | $\lambda_{\mathcal{C}}$ |
| 2-Moon2 | $3 \times 10^{-3}$ | 8 | 8 | 16 | 20.0 | $9 \times 10^{-3}$ | 64 | 3 | 72 | 3 | 1.0 |
| Elec2 | $1 \times 10^{-4}$ | 16 | 8 | 16 | 0.0 | $2 \times 10^{-2}$ | 64 | 2 | 64 | 3 | 2.0 |
| ONP | $1 \times 10^{-4}$ | 10 | 8 | 8 | 5.0 | $1 \times 10^{-2}$ | 72 | 3 | 72 | 3 | 20.0 |
| Shuttle | $1 \times 10^{-4}$ | 10 | 8 | 8 | 1.0 | $1 \times 10^{-2}$ | 64 | 2 | 64 | 2 | 0.1 |
| Appliance | $5 \times 10^{-3}$ | 8 | 8 | 8 | 20.0 | $5 \times 10^{-3}$ | 72 | 3 | 72 | 3 | 1.0 |

LightGBM[4], and FT-Transformer[5]) are detailed below. Note that for MLPs, we use the same structures presented in DRAIN (Bai et al., 2022) to maintain a fair comparison.

**Rotated 2 Moons (2-Moons).** The MLP is structured with 2 hidden layers, each having a dimension of 50. Following each hidden layer, we incorporate a ReLU activation function, and after the output layer, we apply a Sigmoid activation function. The learning rate is set to be $1 \times 10^{-2}$. For the LightGBM, the hyperparameters are set as follows: feature fraction: 1.0; bagging fraction: 0.9; bagging frequency: 1; lambda l1: 0.1; lambda l2: 0.1; linear lambda: 0.1; learning rate: $1 \times 10^{-2}$. For the FT-Transformer, the hyperparameters are configured as: input embedding dimension: 64; number of attention heads: 4; number of attention blocks: 4; attention dropout rate: 0.1; batch size: 32; learning rate: $1 \times 10^{-4}$.

**Electrical Demand (Elec2).** The MLP is structured with 2 hidden layers, each having a dimension of 128. Following each hidden layer, we incorporate a ReLU activation function, and after the output layer, we apply a Sigmoid activation function. The learning rate is set to be $1 \times 10^{-4}$. For the LightGBM, the hyperparameters are set as follows: feature fraction: 0.9; bagging fraction: 1.0; bagging frequency: 30; lambda l1: 0.5; lambda l2: 0.9; linear lambda: 0.1; learning rate: $5 \times 10^{-2}$. For the FT-Transformer, the hyperparameters are configured as: input embedding dimension: 128; number of attention heads: 8; number of attention blocks: 8; attention dropout rate: 0.1; batch size: 128; learning rate: $8 \times 10^{-5}$.

**Online News Popularity (ONP).** The MLP is structured with 2 hidden layers, each having a dimension of 20. Following each hidden layer, we incorporate a ReLU activation function, and after the output layer, we apply a Sigmoid activation function. The learning rate is set to be $1 \times 10^{-3}$. For the LightGBM, the hyperparameters are set as follows: feature fraction: 0.8; bagging fraction: 0.9; bagging frequency: 30; lambda l1: 0.1; lambda l2: 0.1; linear lambda: 0.0; learning rate: $5 \times 10^{-2}$. For the FT-Transformer, the hyperparameters are configured as: input embedding dimension: 64; number of attention heads: 8; number of attention blocks: 2; attention dropout rate: 0.1; batch size: 256; learning rate: $1 \times 10^{-4}$.

**Shuttle.** The MLP is structured with 3 hidden layers, each having a dimension of 128. Following each hidden layer, we incorporate a ReLU activation function, and after the output layer, we apply a Sigmoid activation function. The learning rate is set to be $1 \times 10^{-3}$. For the LightGBM, the hyperparameters are set as follows: feature fraction: 1.0; bagging fraction: 1.0; bagging frequency: 5; lambda l1: 0.1; lambda l2: 0.1; linear lambda: 0.0; learning rate: $5 \times 10^{-2}$. For the FT-Transformer, the hyperparameters are configured as: input embedding dimension: 64; number of attention heads: 4; number of attention blocks: 8; attention dropout rate: 0.1; batch size: 128; learning rate: $1 \times 10^{-4}$.

**Appliances Energy Prediction (Appliance).** The MLP is structured with 2 hidden layers, each having a dimension of 128. Following each hidden layer, we incorporate a ReLU activation function, and after the output layer, we don't apply an activation function. The learning rate is set to be $1 \times 10^{-4}$. For the LightGBM, the hyperparameters are set as follows: feature fraction: 1.0; bagging fraction: 0.8; bagging frequency: 10; lambda l1: 0.5; lambda l2: 0.9; linear lambda: 0.0; learning rate: $5 \times 10^{-2}$. For the FT-Transformer, the hyperparameters are configured as: input embedding dimension: 64; number of attention heads: 4; number of attention blocks: 2; attention dropout rate: 0.9; batch size: 128; learning rate: $1 \times 10^{-4}$.

---

[4] https://lightgbm.readthedocs.io/en/stable/
[5] https://pytorch-tabular.readthedocs.io/en/latest/

# E  QUALITY ANALYSIS VIA VISUALIZATION

To investigate the quality of the CODA-generated data, we further provide the data visualizations in Figure 6 to Figure 8 for comparing the simulated datasets with the ground truth unseen future domains and the data generated by our preliminary setting (see Section 3.1). As shown in Figure 6, the Prelim-LSTM can hardly capture the underlying temporal trends from chronological source domains to accurately predict the future data distribution; on the other hand, the distribution of the CODA-generated data in Figure 6-(c) is similar to the ground truth distribution $\mathcal{D}_9$ in Figure 6-(a). In the Elec2 dataset, Figure 7-(a) demonstrates the impressively closed data distributions between the ground truth and the data generated by CODA; in contrast, the distribution of the data generated by Prelim-LSTM is quite different to the ground truth, as we can see in Figure 7-(b). As the chronological source domains do not manifest strong concept drifts, such as the ONP dataset, both the CODA and Prelim-LSTM cannot precisely simulate the data distribution in the near-future domain, as shown in Figure 8.

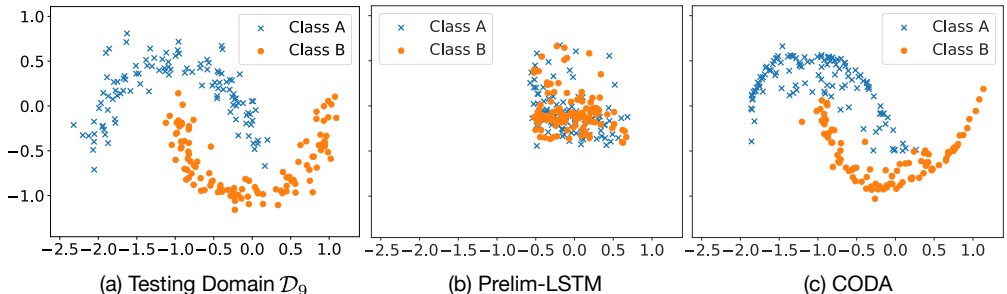

(a) Testing Domain $\mathcal{D}_9$  (b) Prelim-LSTM  (c) CODA

Figure 6: Visualization of generated future domain data in 2-Moons dataset: (a) Groundtruth of test domain data $\mathcal{D}_9$; (b) the synthetic data generated by Prelim-LSTM; (c) the synthetic data $\hat{\mathcal{D}}_9$ generated by CODA.

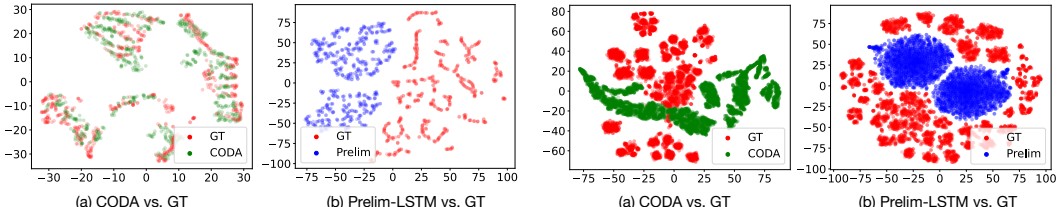

(a) CODA vs. GT  (b) Prelim-LSTM vs. GT    (a) CODA vs. GT  (b) Prelim-LSTM vs. GT

Figure 7: Visual comparison with the ground truth future data (Elec2).

Figure 8: Visual comparison with the ground truth future data (ONP).

# F  VISUALIZATION OF FEATURE CORRELATION MATRIX

To delve deeper into whether the feature correlation matrices reflect the temporal shifts between distinct time points, we visualize the actual feature correlation matrices in Figure 9. We observe that the inter-feature correlations do change over time. Intriguingly, for the regression dataset, Appliance, the prediction target (represented by the last column in the correlation matrices) exhibits pronounced correlations with nearly all features, reflecting the nature of the regression task.

Figure 10. To demonstrate the efficacy of the Correlation Predictor in forecasting the near-future correlation matrix, we present the error maps depicting the absolute difference between $\hat{]}_{T+1}$ and $\mathcal{C}_{T+1}$ in Figure 10. As we can see, except for the ONP dataset, which does not have strong concept drifts over chronological source domains, the Correlation Predictor can overall predict the future feature correlation matrices with decent error levels. However, we can observe that a few cells in correlation matrices still cannot be precisely predicted. The results unveil the potential future direction for further improving the performance of feature correlation matrices prediction.

Figure 9: Visualization of feature correlation matrices from $T-2$ to $T+1$.

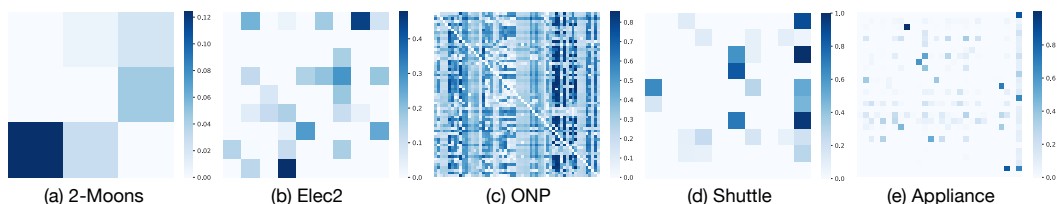

Figure 10: Visualization of the absolute errors between $\hat{\mathcal{C}}_{T+1}$ and $\mathcal{C}_{T+1}$.

# G EXPERIMENTS OF CYCLICAL DISTRIBUTION SHIFT AND HIGH-DIMENSIONAL DATASETS

Although we have already considered diverse concept drift patterns, here we add two more types of concept shift and data, as shown in Table 6. The considered concept shifts are as follows: **Synthetic concept drifts:**

- Cyclical change: the 2-Moons dataset is built with a cyclical concept drift pattern, and we conduct **Rot-MNIST** and **Sine** datasets as shown in the table above.
- Abrupt change: this type of temporal trend doesn't fit our assumption that **"the joint distribution of features and labels with smooth data shift over time"** (refer to Introduction).

**Real-world concept drift:** The real-world datasets used in our experiments feature various and unknown patterns of concept drift. They have covered diverse realistic temporal trends, such as electricity demand changes (Elec2), space shuttle defects (Shuttle), and appliances energy usage changes (Appliance).

Table 6: Performance comparisons on Sine, Rot-MNIST, Portraits, and Forest Cover datasets.

| Frameworks | Sine | Rot-MNIST | Portraits | Forest Cover |
|---|---|---|---|---|
| LSSAE (Qin et al., 2022) | $36.8 \pm 1.5$ | $16.6 \pm 0.7$ | $6.9 \pm 0.3$ | $36.8 \pm 0.4$ |
| DDA (Zeng et al., 2023) | $1.6 \pm 0.9$ | $13.8 \pm 0.3$ | $5.1 \pm 0.1$ | $34.7 \pm 0.5$ |
| GI (Nasery et al., 2021) | $33.2 \pm 0.7$ | $7.7 \pm 1.3$ | $6.3 \pm 0.2$ | $36.4 \pm 0.4$ |
| DRAIN (Bai et al., 2022) | $3.0 \pm 1.0$ | $7.5 \pm 1.1$ | $- \pm -$ | $- \pm -$ |
| CODA (MLP) | $\mathbf{2.7 \pm 0.9}$ | $\mathbf{6.0 \pm 1.2}$ | $5.1 \pm 0.1$ | $\mathbf{34.4 \pm 0.4}$ |
| CODA (LightGBM) | $\mathbf{1.2 \pm 0.4}$ | $\mathbf{5.8 \pm 0.6}$ | $6.2 \pm 0.1$ | $33.0 \pm 0.3$ |
| CODA (FT-Transformer) | $\mathbf{1.1 \pm 0.4}$ | $\mathbf{6.3 \pm 0.5}$ | $\mathbf{4.9 \pm 0.2}$ | $33.7 \pm 0.3$ |

# H TRAINING TIME COMPARISONS

In this work, we mainly focus on the effectiveness of achieving TDG rather than on efficiency.

Although efficiency is not our main goal, we would like to justify that our proposed framework achieves decent efficiency compared to the SOTA method DRAIN.

The reason is that by splitting the whole temporal trend modeling and data generation process into three sub-processes (learning Correlation Predictor $H(\cdot)$, learning Data Simulator $G(\cdot)$, and predictor training), each of the sub-processes is a manageable sub-problem and takes less training time than a whole end-to-end model. We demonstrate the training time comparison to the SOTA DRAIN on Elec2 dataset in Table7, where we train the same MLP structure as the predictor.

Table 7: Performance comparisons on Sine and Rot-MNIST datasets.

| Framework & Components | Training Times (s) |
|---|---|
| DRAIN (Bai et al., 2022) | 465.936 |
| CODA (Total) | 447.817 |
| CODA (Correlation Predictor $\boldsymbol{H}(\cdot)$) | 142.110 |
| CODA (Data Simulator $\boldsymbol{G}(\cdot)$) | 290.826 |
| CODA (MLP) | 14.880 |

