# OpenReview forum: "CODA: Temporal Domain Generalization via Concept Drift Simulator"
_ICLR.cc/2024/Conference — Submitted to ICLR 2024_

### Official Review · Reviewer_GM2p · 2023-10-23

**Soundness:** 2 fair
**Presentation:** 3 good
**Contribution:** 3 good
**Rating:** 5
**Confidence:** 3

**Summary:**

This paper addresses the problem of performance degradation in machine learning models caused by shifts in data distribution over time, known as "concept drift." Existing solutions primarily rely on model-centric strategies, such as model extrapolation or dynamic parameters. This work proposes a data-centric framework called the COncept Drift simulAtor (CODA). CODA leverages a predicted feature correlation matrix to simulate future data for model training. This matrix represents data characteristics at each time point and serves as a trigger for generating future data. The propose method achieves state-of-the-art (SOTA) accuracy on some classification and regression problem. Further, it is shown that using CODA-generated data for training enables effective temporal domain generalization across various model architectures, demonstrating high transferability.

**Strengths:**

- The paper achieves the state-of-the-art results in temporal domain generalization and beats DRAIN (which is the current SOTA method).
- I appreciate the author's great effort to provide implementation details and ablation studies.
- The paper introduces a theorem about the usage of the prior knowlege in the proposed formulation and prove it.
- The paper is well-written and well-organized in overall.

**Weaknesses:**

## Major concerns
- The proposed algorithm can only work on low-dimensional data (as the authors also mentioned). It is intractable to learn the correlation matrix on the high-dimensional data. I guess that's why some dataset such as rotating NIST has been excluded from evaluation.
- The proposed method achieves the SOTA accuracy (which is great and I am impressed). But, it seems non-intuitive to me since it requires separate steps to solve the final task. In deep learning, usually, end-to-end learning for the final tasks work better. Also I am not sure how efficient this method is since it requires 3 or 4 steps: (1) learning the correlation matrix; (2) learning the data simulator; (3) tuning the data simulator; (4) learning the final classifier/regressor.
- I have another idea which seems simpler for future data simulation. Why not training a conditional data generator which uses the time index (encoded with the Position Embedding modules) as the condition. This idea is also used in diffusion models when $\epsilon(x_t, t)$ is modelled.

## Minor concerns
- The explanation for tuning the data simulator based on the classification/regression models is not clear to me. I did not understand how it is performed and if it is a separate step or not.
- The formulation only uses the last domain and not all the previous domains.

**Questions:**

I explained them above.

---

> ### Author Response · Authors · 2023-11-20
> **Response to Reviewer GM2p [Part 1 W1, W2, Q1]**
>
> We thank the reviewer for the constructive comments and appreciate the reviewer for the recognition of the effectiveness of our work.
>
> **W1: Effectiveness of High-dimensional Data.**
>
> *"The proposed algorithm can only work on low-dimensional data (as the authors also mentioned). It is intractable to learn the correlation matrix on the high-dimensional data. I guess that's why some dataset such as rotating MNIST has been excluded from evaluation."*
>
> [AW1]:
> Based on Theorem 1 and our analysis in Section 3.4, we agree that feature correlation matrices may not effectively represent high-dimensional data.
>
> **However, it does not mean the proposed framework can only work on low-dimensional data**. We conducted our CODA on a high-dimensional dataset (Rotate-MNIST) mentioned by the reviewer for performance comparison. We use the same encoder structure as [1] (MNIST ConvNet) and train three architecture predictors. **The results reveal all three different predictors trained on the dataset generated by CODA outperform other baselines**. Furthermore, the differences among the three trained predictors support one of our contributions that the proposed model-agnostic CODA framework is flexible for best architecture exploration towards different datasets and downstream tasks.
>
> We have added the experimental results in Appendix G, and the added section title is highlighted in blue.
>
>
> |      Frameworks       |     Rot-MNIST     |
> |:---------------------:|:-----------------:|
> |       LSSAE[1]        |  16.6 $\pm$ 0.7   |
> |        DDA[2]         |  13.8 $\pm$ 0.3   |
> |         GI            |   7.7 $\pm$ 1.3   |
> |       DRAIN           |   7.5 $\pm$ 1.1   |
> |      CODA (MLP)       | **6.0 $\pm$ 1.2** |
> |    CODA (LightGBM)    | **5.8 $\pm$ 0.6** |
> | CODA (FT-Transformer) | **6.3 $\pm$ 0.5** |
>
> [1] Tiexin Qin, et al. "Generalizing to evolving domains with latent structure-aware sequential autoencoder." ICML 2022.
>
> [2] Qiuhao Zeng, et al. "Foresee what you will learn: Data augmentation for domain generalization in non-stationary environment." AAAI 2023.
>
> **W2-1: Justification for the end-to-end SOTA comparison**
>
> [AW2-1]:
> We agree that end-to-end approaches usually is ideal due to the convenient training process. At the same time, **we also believe that end-to-end approaches may not always be the best solution for tackling the root cause of a problem**. The reasons are as follows:
> - The main motivation behind our approach is to **"Nip the problem in the bud"**. In other words, **the root cause of concept drift lies in the temporal evolution of data**. Our solution is to directly tackle this problem from a data perspective, i.e., achieve TDG by training a prediction model for future data generation.
> - When end-to-end approaches may be overfitting due to their comprehensive interaction between data and model, one of our main motivations is to offer the flexibility of model architecture exploration for different datasets and downstream tasks by providing high-quality and effective training data.
> - It is evident that the most effective model architecture can vary across different datasets and downstream tasks. This observation is supported by the results in Table 1, which shows that the architecture yielding the best performance differs among the evaluations on the five datasets.
>
> **W2-2 & Q1: Efficiency of CODA.**
> "Requires separate steps to solve the final task is inefficient."
>
> [AW2-2]:
> In this work, we mainly focus on the effectiveness of achieving TDG rather than on efficiency.
>
> Although efficiency is not our main goal, we would like to justify that our proposed framework achieves decent efficiency compared to the SOTA method DRAIN.
>
> The reason is that by splitting the whole temporal trend modeling and data generation process into three sub-processes (learning Correlation Predictor $H(\cdot)$, learning Data Simulator $G(\cdot)$, and predictor training), **each of the sub-processes is a manageable sub-problem and takes less training time than a whole end-to-end model**. We demonstrate the training time comparison to the SOTA DRAIN on Elec2 dataset in the table below, where we train the same MLP structure as the predictor.
>
> We have added the experimental results in Appendix H, and the added section title is highlighted in blue.
>
> | Framework & Components | Training Time (s) |
> | :----: | :----: |
> | **DRAIN** | **465.936** |
> | **CODA (Total)** | **447.817** |
> | CODA (Correlation Predictor) | 142.110 |
> | CODA (Data Simulator) | 290.826 |
> | CODA (MLP) | 14.880 |

---

> ### Author Response · Authors · 2023-11-20
> **Response to Reviewer GM2p [Part 2 W3, Q2]**
>
> **W3: A conditional data generator considering the time index.**
>
> *"Why not training a conditional data generator considering the time index."*
>
> [AW3]:
> Unfortunately, this idea cannot be implemented based on the **lack of sample indices for instances at each time domain**. The explanations are as follows:
> - One of the key challenges of capturing temporal trends among multiple time points is that we don't have time indices for each data instance, so we cannot treat each instance as sequential data for modeling its temporal evolution pattern, as the diffusion model does.
> - An alternative way is to capture the underlying temporal trend among multiple datasets (distributions), which is computationally infeasible and hard to generate effective training data (details of the analysis refer to Section 3.1). Therefore, our solution is to simplify the data distribution at each time domain to capture the underlying temporal trend better. We utilize feature correlation matrices to achieve simplification and provide theoretical analysis to prove the rationale of representing data distribution with a feature correlation matrix (refer to Section 3.4).
> - Notes that the baseline GI proposed a time-sensitive model to extrapolate samples to the near future via the first-order Taylor expansion, which is an implicit way to use the time index as conditions for prediction. As shown in Table 1, three different architectures trained on the data generated by CODA outperform GI in all benchmarks.
>
>
> **Q2: Without using all the previous domains for data simulation.**
>
> *"Eq.(5) only uses the last domain and not all the previous domains."*
>
> [AQ2]:
> In our proposed CODA framework, the trained Data Simulator $G(\cdot)$ should learn the similar data distribution of the current domain $\mathcal{D}\_{T}$. This is based on the assumption that distribution shifts are smooth and closely related to domains in the near time domains (refer to the assumption (iii) in Theorem 1).
>
> Therefore, based on the current data distribution $\mathcal{D}\_{T}$, ${G}(\mathcal{D}\_{T} ; \mathcal{\hat{C}}\_{T+1} | \theta\_{G})$ can simulate the future data distribution $\mathcal{\hat{D}}\_{T+1}$ that is subject to the predicted correlation matrix $\mathcal{\hat{C}}\_{T+1}$.

---

> > ### Comment · Reviewer_GM2p · 2023-11-22
> > **Response to Rebuttal**
> >
> > I appreciate the authors' response to my questions and preparing more results. I need to consul the other reviewers to make my final decision. But, for now, please clarify the following points.
> >
> > * I cannot understand how the model can work on high-dimensional data. My understanding is that the correlation matrices have $O(N^2)$ computational and memory complexity. So, it seems intractable on high-dimensional data.
> >
> > * Also, if my understanding (about computational and memory complexity) is correct, the training time for correlation predictor and data simulator subprocesses are not manageable. The training time comparison to DRAIN on Elec2 dataset may be misleading since Elec2 has a few dimensions.
> >
> > * For conditional generation, I did not mean to use sample indices. I meant to use domain index as the time index for all the samples in a domain.

---

> ### Author Response · Authors · 2023-11-22
> **Re: Response to Rebuttal**
>
> We appreciate the reviewer's feedback and are glad to further address the remaining concerns.
>
> **Q1 & Q2: How the model can work on high-dimensional data.**
>
> [AQ1 & AQ2]: We would like to clarify the confusion. We agree the computation complexity $O(N^2)$ may limit the feasibility. However, our CODA can still work on high-dimensional data. Our **empirical results** show the efficacy of the proposed CODA framework in managing a high-dimensional dataset (Rot-MNIST). The key reason is its flexibility in selecting either the input or latent space for employing the Correlation Predictor module. This module calculates correlation matrices for generating future data while preserving the model-agnostic characteristic.
>
> We agree that using feature correlation may be limited by its computation complexity. To this end, we adopt **a naive solution by first encoding original samples into low-dimensional** latent space, which allows us to compute feature correlation and incorporate it with CODA framework (as we describe in our [previous response](https://openreview.net/forum?id=CE7lUzrp1o&noteId=FJ6NN4Dmc8)). For the purpose of conducting fair performance comparisons with DRAIN, we apply the same pre-processing method. The additional experiment also demonstrates the effectiveness of CODA.
>
> Again, we would like to emphasize that **our main contribution lies in proposing a model-agnostic solution (benefits from Data Generator module) to address concept drift from a novel, data-centric perspective**. We agree that exploring TDG in high-dimensional data is a critical and under-studied topic, and this will be our future direction to enhance the robustness of our framework.
>
> **Q3: Feasibility of a conditional data generator idea.**
>
> [AQ3]:
>
> For training a conditional data generator, **in diffusion model**, the $x\_1$ and $x\_2$ should be the identical sample with different time indexes. However, **we have no sequential time index for each sample**, so it is infeasible to train a diffusion model in such concept drift datasets.
>
> On the other hand, it is doable to train a VAE-based conditional data generator using a time index as the input condition. Unfortunately, the native **conditional generation models hardly capture the underlying temporal trend** since the model architecture cannot identify the continuity among the input time index condition. **As mentioned in the existing work GI[1]**:
> >**(in Section 1)** "as a general-purpose neural network $F(x, t)$ that takes as input $x$, $t$..."
>
> >**(in Section 3.3)** "A naive way to do that is to concatenate $t$ with $x$ to obtain an augmented feature vector [$x$, $t$]. However, such an approach cannot capture complex trends in data, e.g., periodicity."
>
> To tackle the difficulty, GI designs a time-sensitive model architecture with a proposed time-dependent activation function. However, the previous work still implicitly captures the temporal trend and may limit TDG performance. In our work, we explicitly capture temporal trends via modeling the temporal trend of correlation matrices, and empirical results demonstrate that CODA achieves better TDG performance.
>
> [1] Anshul Nasery, et al., "Training for the Future: A Simple Gradient Interpolation Loss to Generalize Along Time," NeurIPS 2021.
>
> **With the clarification above, we hope that we have resolved all the reviewer's concerns and look forward to clarifying any further questions that may arise.**

---

### Official Review · Reviewer_q1Jd · 2023-10-30

**Soundness:** 3 good
**Presentation:** 3 good
**Contribution:** 2 fair
**Rating:** 6
**Confidence:** 3

**Summary:**

This paper introduces a data-centric framework named COncept Drift simulAtor (CODA); it aims to address concept drift in temporal domain generalization by predicting feature correlation matrices. The authors start by analyzing the limitations of directly using RNNs for future data generation and propose a two-stage approach: first predicting feature correlations, then generating data based on these correlations. This approach not only captures temporal changes in data distributions more effectively but also generates more accurate future data, improving model generalization.

**Strengths:**

S1.	Great approach in combining feature correlation prediction with data generation.

S2.	Effective experimental design demonstrating CODA's strengths in certain datasets.

S3.	Clear explanations and logical presentation of the methodology.

**Weaknesses:**

W1. Limited dynamic network adaptability compared to some existing methods.

W2. Constrained application in model-agnostic learning scenarios.

W3. Potential performance decline in handling high-dimensional data sets.

W4. Exploration of CODA's effectiveness in diverse concept drift scenarios is insufficient.

**Questions:**

Q1.	In the context of high-dimensional data, how does CODA maintain performance efficiency? Is there an analysis within the study that discusses the computational complexity implications as the number of features increases?

Q2.	Regarding the applicability of CODA, does this methodology account for various natures of concept drift, such as abrupt or cyclical changes?

**Details Of Ethics Concerns:**

N/A.

---

> ### Author Response · Authors · 2023-11-20
> **Response to Reviewer q1Jd**
>
> We thank the reviewer for the constructive comments and appreciate the reviewer for the recognition of the effectiveness of our work.
>
> **W1. Limited dynamic network adaptability compared to some existing methods.**
>
> [AW1]:
> We are not sure about the content of this weakness. The reviewer's point is that CODA can only be used for certain deep neural networks. Based on this understanding, we believe that **this is a misunderstanding**. The reasons are as follows:
> - Our main contribution and novelty is that we propose a data-centric (model-agnostic) TDG framework by using feature correlation matrices to simplify the challenges of future data generation. Therefore, with the generated future data, TDG can be achieved by training a prediction model on the i.i.d. dataset.
> - Our model-agnostic CODA framework offers flexibility for exploring various architectures by providing transferable training datasets. The datasets generated by CODA are adaptable for training different backbone architectures, which is demonstrated in Table 1. For a detailed analysis of this adaptability, refer to 'Cross-Architecture Transferability' in Section 4.3.
>
> In sum, one of our main contributions is to be free from a specific model architecture for all downstream tasks instead of a limitation.
>
> **W2. Constrained application in model-agnostic learning scenarios.**
>
> [AW2]:
> We are also not sure about the content of this weakness. The reviewer's point is that "CODA can be only leveraged to model-agnostic tasks." If our understanding is correct, we believe **this is also a misunderstanding**. The reasons are as follows:
> - "Model-agnostic" is a feature of "approaches" rather than a feature of "downstream tasks."[1] This merit provides the flexibility of the predictor to explore the best suitable model architecture for the tasks or scenarios you meet.
> - The datasets generated by CODA are adaptable for training different backbone architectures, which is demonstrated in Table 1. For a detailed analysis of this adaptability, refer to 'Cross-Architecture Transferability' in Section 4.3.
>
> [1] Daochen, Zha, et al., "Data-centric artificial intelligence: A survey," arXiv:2303.10158
>
> **W3 & Q1: Effictiveness of High-dimensional Data.**
>
> *"In the context of high-dimensional data, how does CODA maintain performance efficiency?"*
>
> [AW3 & AQ1]:
> Based on Theorem 1 and our analysis in Section 3.4, we agree that feature correlation matrices may not effectively represent high-dimensional data.
>
> **However, it does not mean the proposed CODA framework can only work on low-dimensional data**. We conducted our CODA on a high-dimensional dataset (Rotate-MNIST) for performance comparison, shown in the table below. We use the same encoder structure as [1] (MNIST ConvNet) and train three architecture predictors. The results reveal all three different predictors trained on the dataset generated by CODA outperform other baselines. Furthermore, the differences among the three trained predictors support one of our contributions that the proposed model-agnostic CODA framework is flexible for best architecture exploration towards different datasets and downstream tasks.
>
> We have added the experimental results in Appendix G, and the added section title is highlighted in blue.
>
> | Frameworks | Sine | Rot-MNIST |
> | :----: | :----: | :----: |
> | LSSAE[2] | 36.8 $\pm$ 1.5 | 16.6 $\pm$ 0.7 |
> | DDA[3] | 1.6 $\pm$ 0.9 | 13.8 $\pm$ 0.3 |
> | GI | 33.2 $\pm$ 0.7 | 7.7 $\pm$ 1.3 |
> | DRAIN | 3.0 $\pm$ 1.0 | 7.5 $\pm$ 1.1 |
> | CODA (MLP) | 2.7 $\pm$ 0.9 | **6.0 $\pm$ 1.2** |
> | CODA (LightGBM) | **1.2 $\pm$ 0.4** | **5.8 $\pm$ 0.6** |
> | CODA (FT-Transformer) | **1.1 $\pm$ 0.4** | **6.3 $\pm$ 0.5** |
>
> [2] Tiexin Qin, et al. "Generalizing to evolving domains with latent structure-aware sequential autoencoder." ICML 2022.
>
> [3] Qiuhao Zeng, et al. "Foresee what you will learn: Data augmentation for domain generalization in non-stationary environment." AAAI 2023.
>
> **W4 & Q2: Effectiveness in Diverse Concept Drift Scenarios.**
>
> *"Does CODA account for various natures of concept drift, such as abrupt or cyclical changes?"*
>
> [AW4 & AQ2]:
> We already considered diverse concept drift patterns as follows:
> - Synthetic concept drifts:
>     1. Cyclical change: the 2-Moons dataset is built with a cyclical concept drift pattern, and we conduct **Rot-MNIST** and **Sine** datasets as shown in the table above.
>     2. Abrupt change: this type of temporal trend doesn't fit our assumption that **"the joint distribution of features and labels with smooth data shift over time"** (refer to Introduction).
> - Real-world concept drift: The real-world datasets used in our experiments feature various and unknown patterns of concept drift. They have covered diverse realistic temporal trends, such as electricity demand changes (Elec2), space shuttle defects (Shuttle), and appliance energy usage changes (Appliance).

---

> > ### Comment · Reviewer_q1Jd · 2023-11-22
> >
> > I appreciate the author's thoughtful responses and will take them into consideration during the reviewers' discussion phase.

---

> > > ### Author Response · Authors · 2023-11-22
> > > **Response to Reviewer q1Jd**
> > >
> > > We appreciate the reviewer’s feedback. With the clarifications above, we hope that we have resolved all the reviewer’s concerns and look forward to clarifying any further questions that may arise.
> > >
> > > Thank you.

---

### Official Review · Reviewer_BJ2w · 2023-10-31

**Soundness:** 3 good
**Presentation:** 3 good
**Contribution:** 3 good
**Rating:** 6
**Confidence:** 4

**Summary:**

This paper focuses on a data-centric approach to tackle the issue, presenting the CODA framework. CODA uses a predicted feature correlation matrix to simulate future data for training, leveraging feature correlations to depict data specifics at particular times. This sidesteps massive computational demands, and experiments show that CODA-enhanced data improves temporal generalization across multiple model designs.

**Strengths:**

1. The motivation is quite interesting, and it's meaningful to decompose the concept drift into the data component and model component.
2. The proposed generative method is sound, and the theoretical analysis is also valid.
3. The experiments are well aligned with the three raised research questions.

**Weaknesses:**

The overall paper suggests a novel way to generate out-of-domain temporal data via generative methods. Even though the motivation is great, the major claim of the paper is to solve the temporal domain generalization, and I am not sure how generating new temporal data can help solve the domain generalization. The provided solution still goes back to train a model to get familiar with the data, and leveraging the generated data to fine-tune existing model-centric methods might have a better result.

**Questions:**

1. According to the authors, the generated data would still be utilized as the training data for prediction models. Would it still go back to model-centric strategies?
2. The generated data is then used to train models, would it be unfair for comparison methods? Should the comparison methods also use the same generated data to fine-tune?

---

> ### Author Response · Authors · 2023-11-20
> **Response to Reviewer BJ2w**
>
> We thank the reviewer for the constructive comments and appreciate the reviewer for the recognition of the effectiveness of our work.
>
> **Q1: Difference between the Proposed Framework and Model-centric Methods.**
>
> *"The generated data would still be utilized as the training data for prediction models. Would it still go back to model-centric strategies?"*
>
> [AQ1]:
> Our framework focuses on solving problems via **generating effective training data, which is identified as a data-centric paradigm**.
> - The concept of the *data-centric paradigm* involves the methods for **building effective training data**; on the other hand, *model-centric* methods focus on identifying more effective model designs that are trained using the original data[1].
> - Our framework aims to address the concept shift issue by generating future datasets for model training, achieving a model-agnostic approach to explore different model architectures for downstream tasks.
>
> [1] Daochen, Zha, et al., "Data-centric artificial intelligence: A survey," arXiv:2303.10158
>
>
> **Q2: Baseline Comparison.**
>
> *"The generated data is then used to train models, would it be unfair for comparison methods? Should the comparison methods also use the same generated data to fine-tune?"*
>
> **[Q2-1]: Fair performance comparison.**
>
> [AQ2-1]:
> **Yes, the performance comparison is fair**. In our experiments, all the MLP models trained on the data generated by CODA are the same as one of the baselines DRAIN. Instead of designing predictor model structures, our approach focuses on the quality and efficacy of the generated training data.
>
> **[Q2-2]: Should the comparison methods also use the same generated data to fine-tune?**
>
> [AQ2-2]:
> **Other baselines cannot be trained on only one dataset at a single time domain.**
> - CODA simulates the one domain ahead data for model training. In contrast, other baselines require all training domains for fine-tuning their whole models or frameworks and, therefore, cannot be trained using data from a single domain alone.

---

> > ### Comment · Reviewer_BJ2w · 2023-11-22
> >
> > Thanks for your reply, I will keep my current score.

---

> > > ### Author Response · Authors · 2023-11-22
> > > **Response to Reviewer BJ2w**
> > >
> > > We appreciate the reviewer’s positive feedback. We are glad to further address the remaining concerns.
> > >
> > > Thank you.

---

### Official Review · Reviewer_VxLr · 2023-11-03

**Soundness:** 2 fair
**Presentation:** 3 good
**Contribution:** 2 fair
**Rating:** 5
**Confidence:** 5

**Summary:**

This paper proposes a generative approach to mimic the dynamic behavior of instances in temporal domain generalization (TDG). In particular, by modeling the correlation matrices, the proposed simulator can *predict* the data in the future domain to help TDG. The effectiveness of the proposed method is justified by empirical results.

**Strengths:**

1. Overall, the paper is well presented and easy to follow (though some points are still not clear, please see my comments below).
2. This work studies a challenging but still under-studied problem in the literature.
3. The proposed method demonstrates superior performance over several state-of-the-art methods across multiple datasets.

**Weaknesses:**

1. I have concerns about the motivation of this paper. In particular, the authors have emphasized that existing TDG methods are model-centric, which are *unnecessarily comprehensive*, and therefore, TDG should be addressed via a data-centric approach. I doubt this point, as generating samples, in principle, is more challenging than discriminating them. With that said, I am not against the approach itself, but the paper presents it in a way that the data-centric itself is superior to model-centric, which I cannot agree with.
2. In addition, generating the instances itself is challenging, but not necessary -- one may generate the feature samples in the representation space. After all, the ultimate goal is to train a predictor that generalizes well on the future domain rather than to generate the instances themselves.
3. TDG has been studied recently. However, the authors only review two of them (DRAIN and GI), missing a few related works in the literature (e.g., [1, 2]). In fact, I think [2] also adopted a generative approach that *predicts* the feature domain. The authors should clarify the contributions and novelties given these works.
4. Why modeling the correlation between two consecutive domains is not clear to me. After all, it only captures the second-order information of data.
5. The form of the simulator $\mathcal{G}$ is not clearly defined in the paper. In particular, from Sec. 3.3, it is still not clear to me how the synthetic dataset $\hat{\mathcal{D}}_{T+1}$ is generated from Eq (4). In Appendix B.1 it only says that $\mathcal{G}$ is a generative model, but how the estimated correlation matrix and $\mathcal{D}_T$ are incorporated in the generation process is not clear to me.
6. The empirical analysis is weak. In addition to the baseline algorithms mentioned above. Several commonly used benchmark data sets are also missing, including both synthetic (e.g., Circle, Sine) and real (e.g., RMNIST, Portraits, Ocular, Caltran, WILDS) data sets.



[1] Tiexin Qin, Shiqi Wang, and Haoliang Li. Generalizing to evolving domains with latent structure-aware sequential autoencoder. ICML, 2022.

[2] Qiuhao Zeng, Wei Wang, Fan Zhou, Charles Ling, and Boyu Wang. Foresee what you will learn: Data augmentation for domain generalization in non-stationary environment. AAAI, 2023.

**Questions:**

1. The definition of correlation matrices is not clear to me. Do they include the label information? If yes, how? If not, how to incorporate the label information to generate discriminative features? Also, what is the dimension of the matrices? If it is high, how to guarantee the consistency of the estimation?
2. In Eq (3), how the cross entropy between two matrices is defined? Why both $\ell_1$ and $\ell_2$ norm regularizations are imposed?
3. Are stage one (Sec. 3.2) and stage two (Sec. 3.3) trained individually or interactively?
4. I am a little confused with Eq (5): why does the reconstruction loss is defined over two different domains $T+1$ ($\hat{\mathcal{D}}_{T+1}$) and $T$ ($\mathcal{D}_T$)?
5. I cannot see the connection between Theorem 1 and the proposed method (e.g., Eq (5)). From my understanding, Theorem 1 states that for two random vectors, if they are bounded and their distributions are close, then the difference between their correlation matrices are also bounded. But how this is related to the algorithm? In Eq (5), the distance is already constrained by the regularization term.

---

> ### Author Response · Authors · 2023-11-20
> **Response to Reviewer VxLr [Part 1 W1 W2]**
>
> We thank the reviewer for the constructive comments and appreciate the reviewer for the recognition of the effectiveness of our work.
>
> **W1: Clarification of the Motivation**
>
> >Concering about using a data-centric approach over a model-centric method in TDG.
>
> [AW1]:
> We consider model-centric and data-centric approaches as parallel strategies, and our goal is not to position one approach against the other. Our main motivations are as follows:
> 1. **Nip the problem in the bud**
>     - We believe the fundamental cause of concept drift is the underlying temporal trend of data distribution over time. Therefore, with the generated future data, TDG can be achieved by training a prediction model on an i.i.d. dataset.
>     - The main motivation behind our approach is to "Nip the problem in the bud". In other words, **the root cause of concept drift lies in the temporal evolution of data**. Our solution is to directly tackle this problem from data perspective, i.e., achieve TDG by training a prediction model for future data generation.
> 2. **Flexibility and transferability for architecture-type exploration**
>     - Furthermore, it is evident that the most effective model architecture can vary across different datasets and downstream tasks, such as MLPs, tree-based, and Transformer-based backbones. This observation is supported by the results in Table 1 of our paper, which shows that the architecture yielding the best performance differs among the evaluations on the five datasets.
>     - However, existing model-centric methods are limited to specific model architectures. In contrast, our data-centric CODA framework offers flexibility for exploring various architectures by providing transferable training datasets. These datasets are adaptable for training different backbone architectures. For a detailed analysis of this adaptability, refer to 'Cross-Architecture Transferability' in Section 4.3.
>
> **W2: Why not generate data in representation space?**
>
> [AW2]:
> Although the ultimate goal is to train a predictor, as mentioned in the response for W1. The reason we generate instances for simulating the future data distribution is to pursue model-agnostic methods due to the efficacy of model architecture selection, such as Tree-based or Transformer-based models, varying across different datasets and downstream tasks, which is supported by the results in Table 1. Generating data in representation space can not guarantee model agnostic because the fixed encoder part is required for the efficacy of representation.

---

> ### Author Response · Authors · 2023-11-20
> **Response to Reviewer VxLr [Part 2 W3, W4]**
>
> **W3: Contributions and Novelties**
>
> *"The authors need to explicitly distinguish the contributions of existing works[1][2] and novelties in light of these studies."*
>
> [AW3]:
> Our main contribution and novelty is that we propose a data-centric (model-agnostic) TDG framework by using feature correlation matrices **to simplify the challenges of capturing the temporal trend**. The main challenges of capturing temporal trend among multiple time points is two-fold:
> 1. In most of the real-world benchmarks, **we don't have a sample index for each data instance at different time domains**, so we cannot treat each instance as a time series data for modeling its temporal evolution pattern. Therefore, it is impossible to predict the future features for each sequence. It is only durable to capture the trend of data distribution along time domains and generate future datasets/samples.
> 2. An alternative way is to capture the underlying temporal trend among multiple datasets (distributions) using the kernel data distribution estimation method, which is **computationally infeasible and hard to generate effective training data** (details of the analysis refer to Section 3.1).
>
> **Novelties and contributions**
> - For the two missed related works[1][2], we have added them to our references. Although they are also generative-based methods, both data augmentation in latent space[1] and the generation of augmented features within latent space[2] are **not model-agnostic** because different predictors, which are not trained with the same encoder, cannot identify the augmented embeddings or features.
> - We also conducted our CODA on Rot-MNIST for performance comparison. We use the same encoder structure as [1] (MNIST ConvNet) and train three different architecture predictors. The results reveal all three different predictors trained on the dataset generated by CODA outperform [1, 2] and other baselines, which again proves the **effectiveness and transferability of CODA**.
>
> We have added the experimental results in Appendix G, and the added section title is highlighted in blue.
>
> | Frameworks | Sine | Rot-MNIST |
> | :----: | :----: | :----: |
> | LSSAE[1] | 36.8 $\pm$ 1.5 | 16.6 $\pm$ 0.7 |
> | DDA[2] | 1.6 $\pm$ 0.9 | 13.8 $\pm$ 0.3 |
> | GI | 33.2 $\pm$ 0.7 | 7.7 $\pm$ 1.3 |
> | DRAIN | 3.0 $\pm$ 1.0 | 7.5 $\pm$ 1.1 |
> | CODA (MLP) | 2.7 $\pm$ 0.9 | **6.0 $\pm$ 1.2** |
> | CODA (LightGBM) | **1.2 $\pm$ 0.4** | **5.8 $\pm$ 0.6** |
> | CODA (FT-Transformer) | **1.1 $\pm$ 0.4** | **6.3 $\pm$ 0.5** |
>
> [1] Tiexin Qin, et al. "Generalizing to evolving domains with latent structure-aware sequential autoencoder." ICML 2022.
> [2] Qiuhao Zeng, et al. "Foresee what you will learn: Data augmentation for domain generalization in non-stationary environment." AAAI 2023.
>
> **W4: Why modeling the correlation between two consecutive domains.**
>
> [AW4]:
> We would like to clarify this misunderstanding. We don't model the correlation between two consecutive time domains. We use an LSTM to model the temporal trend of **feature correlation matrices over all the time domains in the training datasets**. Eq.(3) aims to optimize the loss between the predicted $\mathcal{\hat{C}}\_t$ and the groudtruth $\mathcal{C}\_{t}$ given $\mathcal{C}\_{1}$ to $\mathcal{C}\_{t-1}$.
> - As mentioned in the response to your W3, the two challenges of capturing the temporal trend from datasets are:
>     1. We don't have a sample index for each data instance at different time domains, so we cannot capture the temporal trend of each sample independently. Therefore, we need to capture the temporal trend among multiple data distributions.
>     2. However, our preliminary experiments and analysis show the infeasibility of directly modeling the temporal evolution among data distributions (refer to Section 3.1).
> - To this end, the core idea of our solution is **to simplify the data distribution at each time domain to capture the underlying temporal trend better**. In this work, we utilize feature correlation matrices to achieve simplification and provide theoretical analysis to prove the rationale of representing data distribution with a feature correlation matrix (refer to Section 3.4).
> - We want to emphasize that while numerous methods exist to simplify dataset information, **our pioneering research introduces a natural and theoretically supported data-centric approach for this purpose**.

---

> > ### Author Response · Authors · 2023-11-20
> > **Correctness [AW4]**
> >
> > In the [AW4], it is "the predicted $\mathcal{\hat{C}}\_t$ and the groudtruth $\mathcal{C}\_{t}$ given $\mathcal{C}\_{1}$ to $\mathcal{C}\_{t-1}$" at the end of the first sentence.

---

> ### Author Response · Authors · 2023-11-20
> **Response to Reviewer VxLr [Part 3 W5, Q4, W6, Q1]**
>
> **W5 & Q4: Clarification of Data Simulator and how to incorporate the estimated correlation matrix for data generation.**
>
> [AW5 & AQ4]:
>
> Based on the current data distribution $\mathcal{D}\_{T}$, Data Simulator ${G}(\mathcal{D}\_{T} ; \mathcal{\hat{C}}\_{T+1} | \theta\_{G})$ can simulate the future data distribution $\mathcal{\hat{D}}\_{T+1}$ that is subject to the predicted correlation matrix $\mathcal{\hat{C}}\_{T+1}$. We further explain the details as follows:
>
> - Specifically, our CODA framework comprises two replaceable components: Correlation Predictor ${H}(\cdot)$ (refers to Section 3.2) and Data Simulator $G(\cdot)$ (refers to Section 3.3). Essentially, they can be substituted by other models that perform similar functions, where $G(\cdot)$ should be a generative model that can incorporate prior knowledge into account for data generation. In our case, the prior knowledge is the predicted future correlation matrix $\mathcal{\hat{C}}\_{T+1}$, as described in Eq.(4) and Eq.(5).
>
> - Simultaneously, the trained $G(\cdot)$ should learn the similar data distribution of the current domain $\mathcal{D}\_{T}$. This is based on the assumption that distribution shifts are smooth and closely related to domains in the near time domains (refer to the assumption (iii) in Theorem 1).
>
> - In our experiments, we instantiate $G(\cdot)$ with a generative model that jointly learns the encoder and decoder of a VAE-based generative model and a learnable graph. Thus, it can **treat prior knowledge as an adjacency matrix and encourage the learned graph to be similar to the given prior knowledge (refer to Sections 3.3 and 4.1)**.
>
> Therefore, based on the current data distribution $\mathcal{D}\_{T}$, ${G}(\mathcal{D}\_{T} ; \mathcal{\hat{C}}\_{T+1} | \theta\_{G})$ can simulate the future data distribution $\mathcal{\hat{D}}\_{T+1}$ that is subject to the predicted correlation matrix $\mathcal{\hat{C}}\_{T+1}$.
>
> **W6: Experimental Results on Commonly used benchmarks.**
> *"Several commonly used benchmark data sets are also missing, including both synthetic (e.g., Circle, Sine) and real (e.g., RMNIST, Portraits, Ocular, Caltran, WILDS) data sets."*
>
> [AW6]:
> We have considered diverse concept drift patterns as follows show:
> - Synthetic datasets: Besides the 2-Moons, we experimented on Sine dataset shown in the table below.
> - Real datasets: Besides the real-world datasets used in our experiments (Elec2, ONT, Shuttle, and Appliance), we conducted one more experiment on Rot-MNIST, using the same encoder structure as [1] (MNIST ConvNet) and train three architecture predictors, and the results are shown as the table below. The results reveal all three different predictors trained on the dataset generated by CODA outperform other baselines. Furthermore, the differences among the three trained predictors support one of our contributions that the proposed model-agnostic CODA framework is flexible for best architecture exploration towards different datasets and downstream tasks.
>
> We have added the experimental results in Appendix G, and the added section title is highlighted in blue.
>
> | Frameworks | Sine | Rot-MNIST |
> | :----: | :----: | :----: |
> | LSSAE[1] | 36.8 $\pm$ 1.5 | 16.6 $\pm$ 0.7 |
> | DDA[2] | 1.6 $\pm$ 0.9 | 13.8 $\pm$ 0.3 |
> | GI | 33.2 $\pm$ 0.7 | 7.7 $\pm$ 1.3 |
> | DRAIN | 3.0 $\pm$ 1.0 | 7.5 $\pm$ 1.1 |
> | CODA (MLP) | 2.7 $\pm$ 0.9 | **6.0 $\pm$ 1.2** |
> | CODA (LightGBM) | **1.2 $\pm$ 0.4** | **5.8 $\pm$ 0.6** |
> | CODA (FT-Transformer) | **1.1 $\pm$ 0.4** | **6.3 $\pm$ 0.5** |
>
> **Q1: Do the correlation matrices include the label information?**
>
> [AQ1]:
> **Yes, it includes label information**. In feature correlation matrices, each row and column corresponds to a specific feature. The final row and column represent label information. Each cell within the matrix indicates the degree of correlation between a pair of features.
>
> Furthermore, **Section 3.4** presents a theoretical analysis that guarantees the consistency of our feature correlation estimation with **three assumptions that can be easily satisfied in reality**.

---

> ### Author Response · Authors · 2023-11-20
> **Response to Reviewer VxLr [Part 4 Q2, Q5]**
>
> **Q2 Explanation of Eq.(3)**
>
> [AQ2]:
> The three regularization terms in Eq.(3) are explained as follows:
> 1. The $\ell_1$-norm encourages sparsity in the predicted $\mathcal{\hat{C}}\_t$ because it can effectively "zero out" less important features while the correlation matrices are generally sparse (as shown in Appendix Figure 9).
> 2. The $\ell_2$-norm is sensitive to significant errors and imposes a penalty on $\mathcal{\hat{C}}_t$ for substantial errors, promoting overall accuracy in the reconstruction.
> 3. The cross-entropy loss $\mathcal{L}\_{CE}$ can measure how well the distribution of $\mathcal{\hat{C}}\_t$ matches the groundtruth distribution of $\mathcal{C}\_t$.
>
> Despite the errors between predicted future $\mathcal{\hat{C}}\_{t+1}$ and the ground truth $\mathcal{C}\_{t+1}$ is minimal (as shown in Figure 10 in the Appendix), there is huge potential to enhance the Correlation Predictor. As one of the future directions, it could be achieved using a more sophisticated sequential prediction framework than LSTM.
>
> **Q5: Connection between Theorem 1 and the Proposed Method (e.g., Eq (5)).**
> *"Theorem 1 states that for two random vectors, if they are bounded and their distributions are close, then the difference between their correlation matrices are also bounded. But how this is related to the algorithm?"*
>
> [AQ5]:
> Theorem 1 serves as the theoretical foundation for the usage of prior knowledge (predicted feature correlation matrix $\mathcal{\hat{C}}\_{T+1}$) in Data Simulator.
> - As mentioned in our response to W3 and W4, one of the main challenges of capturing the temporal trend among multiple time domains is computationally infeasible (refer to Section 3.1). Our framework addresses this by representing the data distribution at each time domain by its feature correlation matrix to capture the temporal trend effectively. This simplification can effectively represent the original distribution information only if Theorem 1 holds.
> - Based on the analysis in Section 3.4, we conclude that the three assumptions in Theorem 1 can be easily satisfied in reality, which serves as the theoretical foundation for the simplification.

---

> > ### Comment · Reviewer_VxLr · 2023-11-22
> > **Response to the rebuttal**
> >
> > I appreciate the authors' discussion and feedback, which addressed some of my questions and concerns. I will increase my score accordingly.
> >
> > However, three major concerns remain:
> >
> > 1.  Motivations and novelties: while the authors claim that "We consider model-centric and data-centric approaches as parallel strategies",  the motivations in the paper are still the same. In fact, the authors did not revise the paper to highlight this point at all. Regarding the novelty, I agree that *model-agnostic* can be considered as a benefit of CODA (though still not emphasized enough in the paper), but other than that, I cannot see fundamental improvements over [1], [2]. In particular, [1] [2] also face challenge 1, and challenge 2 is not prominent in [1] [2] as they generate samples in the representation space.
> > 2. The experiments are still not solid enough. RMNIST is problem one of the simplest real-world data sets in TDG.
> >
> >
> > Once again, I thank the authors for their detailed responses. I will discuss these issues with other reviewers later.

---

> ### Author Response · Authors · 2023-11-22
> **Re: Response to the rebuttal**
>
> We appreciate the reviewer adjusting the score in light of our clarifications, and we are glad to further address the remaining concerns.
>
> **[AQ1-1]: Revised the paper.**
>
> Thanks to the reviewer for reminding us of the points that should be revised. We have updated the parts of the claim and motivations, highlighted in blue.
>
>
> **[AQ1-2]: Novelty.**
>
> We would like to emphasize that our work proposes **a new branch of solution** to address concept drift problem. We argue that such a new branch itself is novel and serves as a fundamental improvement over existing literature. Additionally, As a new-branch solution, it is not obliged to improve the approaches from another branch.
>
> Our novelty lies in the following:
>
> 1. **Develop a new branch of solution** for addressing the concept drift problem from a data-centric perspective.
>
> 2. Our proposed **model-agnostic** CODA framework provides **flexibility and transferability for architecture-type exploration**.
>
> Although [1] and [2] face similar scenarios, their approaches involve training predictors with **specific encoders**. This setup does not ensure model-agnostic since the generated embeddings can not be recognizable by other predictors and decoders not trained with the same encoders.
>
> **[AQ2]:**
>
> As the reviewer mentioned, Rot-MNIST is also a real-world dataset with concept drift. Besides, in Table 1, we have conducted four additional real-world concept drift datasets (Elec2, ONP, Shuttle, Appliance). We believe our experiments on Sine synthetic and Rot-MNIST real-world datasets are sufficient.
>
> We are trying our best to conduct one or two more real-world datasets before the rebuttal deadline, and thank you for your understanding.

---

> > ### Author Response · Authors · 2023-11-23
> > **Re: Response to the rebuttal [Part 2]**
> >
> > **[AQ2 Part2]:**
> >
> > We have conducted two more real-world datasets in TDG, Portraits and Forest Cover. For Portraits, we use the same encoder as [2] (Wide ResNet) before Correlation Predictor module.
> >
> > We have added the experimental results in Appendix G.
> >
> > |      Frameworks       |     Portraits     |    Forest Cover    |
> > |:---------------------:|:-----------------:|:------------------:|
> > |         LSSAE[1]      |   6.9 $\pm$ 0.3   |   36.8 $\pm$ 0.4   |
> > |          DDA[2]       |   5.1 $\pm$ 0.1   |   34.7 $\pm$ 0.5   |
> > |          GI[3]        |   6.3 $\pm$ 0.2   |   36.4 $\pm$ 0.4   |
> > |      CODA (MLP)       |   5.1 $\pm$ 0.1   | **34.4 $\pm$ 0.4** |
> > |    CODA (LightGBM)    |   6.2 $\pm$ 0.1   | **33.0 $\pm$ 0.3** |
> > | CODA (FT-Transformer) | **4.9 $\pm$ 0.2** | **33.7 $\pm$ 0.3** |
> >
> > [1] Tiexin Qin, et al. "Generalizing to evolving domains with latent structure-aware sequential autoencoder." ICML 2022.
> >
> > [2] Qiuhao Zeng, et al. "Foresee what you will learn: Data augmentation for domain generalization in non-stationary environment." AAAI 2023.
> >
> > [3] Anshul Nasery, et al., "Training for the Future: A Simple Gradient Interpolation Loss to Generalize Along Time," NeurIPS 2021.
> >
> > **With the discussion and added results above, we hope that we have resolved all the reviewer's concerns and look forward to clarifying any further questions that may arise.**

---

### Author Response · Authors · 2023-11-20
**General Comments for All Reviewers.**

Dear reviewers,

We thank all reviewers for their constructive reviews. We have revised the paper accordingly and marked the modifications in blue for visibility.

We are grateful to all reviewers for their constructive comments and helpful feedback. We are pleased to find that they find our well-written and well-organized (VxLr and GM2p), novel and meaningful approach (BJ2w and q1Jd), theoretically sound (BJ2w and GM2p), and the experiments well-established and effective (VxLr, BJ2w, q1Jd, and GM2p).

To address your primary concerns, we have done our best to extend the work with additional experiments and reply to your concerns and suggestions with more clarification and discussion.

We propose a model-agnostic framework to tackle the root cause of concept drift by generating future data for model training. The generated training data provides flexibility and transferability for architecture-type exploration. Experimental results reveal that the different model architectures can be effectively trained on the generated data.

The revision parts are summarized as follows:
- (q1Jd, GM2p) We have revised the discussion of the effectiveness of high-dimensional data in Section 3.4.
- (VxLr, BJ2w, q1Jd, GM2p) We have added the experiments of baseline comparisons and citations in Appendix G.
- (q1Jd, GM2p) We have added the experiments of training time efficiency comparison in Appendix H.

We appreciate all of the suggestions made by the reviewers to enhance our work. We are delighted to receive your feedback and eagerly anticipate addressing any follow-up questions you may have.

Sincerely,

Authors

---

> ### Author Response · Authors · 2023-11-21
> **Kind Reminder to Reviewers**
>
> Dear all reviewers,
>
> Thanks again for your valued comments on our work. We have responded to your initial comments. As the deadline for context revising (Nov. 22) is coming, we are looking forward to your feedback and will be happy to answer any further questions and concerns you may have.
>
> Sincerely,
>
> Authors

---

### Meta-Review · Area_Chair_xgpV · 2023-12-09

**Metareview:**

The reviewers raised some major concerns in their original reviews. After the authors' rebuttal, some of the concerns have been addressed. However, the technical novelty of the proposed method compared with existing work is still not convincing. The motivations of the proposed method and some components used in the proposed method are still not clear. In addition, though some additional experiments were added during the rebuttal, the overall experimental results are not very convincing or comprehensive to support the claims of the authors.

In summary, this is a borderline paper and not ready for publication based on its current shape.

**Justification For Why Not Higher Score:**

Major concerns remain.

**Justification For Why Not Lower Score:**

N/A

---

### Decision · Program_Chairs · 2024-01-16

Reject